# A Bound on Thermal Relativistic Correlators at Large Spacelike Momenta

**Souvik Banerjee,**[a] **Kyriakos Papadodimas,**[b] **Suvrat Raju,**[c] **Prasant Samantray**[d] **and Pushkal Shrivastava**[c]

[a]*Department of Physics and Astronomy, Uppsala University, SE-751 08 Uppsala, Sweden*

[b]*International Centre for Theoretical Physics, Strada Costiera 11, 34151 Trieste, Italy*

[c]*International Centre for Theoretical Sciences, Tata Institute of Fundamental Research, Shivakote, Bengaluru 560089, India.*

[d]*Department of Physics, BITS-Pilani Hyderabad Campus, Jawahar Nagar, Shamirpet Mandal, Secunderabad 500078, India*

*E-mail:* souvik.banerjee@physics.uu.se, kyriakos@ictp.it, suvrat@icts.res.in, prshkumar@gmail.com, pushkal.shrivastava@icts.res.in

ABSTRACT: We consider thermal Wightman correlators in a relativistic quantum field theory in the limit where the spatial momenta of the insertions become large while their frequencies stay fixed. We show that, in this limit, the size of these correlators is bounded by $e^{-\beta R}$, where $R$ is the radius of the smallest sphere that contains the polygon formed by the momenta. We show that perturbative quantum field theories can saturate this bound through suitably high-order loop diagrams. We also consider holographic theories in $d$-spacetime dimensions, where we show that the leading two-point function of generalized free-fields saturates the bound in $d = 2$ and is *below* the bound for $d > 2$. We briefly discuss interactions in holographic theories and conclude with a discussion of several open problems.

## 1  Introduction

In this paper, we consider a novel limit of thermal correlation functions in a relativistic quantum field theory in $d$ spacetime dimensions. Let $\mathcal{O}(t, \vec{x})$ be any local operator, which may be the elementary field itself or a more complicated operator. Then consider the Fourier transformed correlation function at finite temperature $\frac{1}{\beta}$

$$\mathcal{W}(\omega_i, \vec{k}_i)\delta_\omega \delta_{\vec{k}} \equiv \int \prod_i dt_i d\vec{x}_i \frac{1}{Z} \mathrm{Tr}\left[ e^{-\beta H} \mathcal{O}(t_1, \vec{x}_1) \dots \mathcal{O}(t_n, \vec{x}_n) \right] e^{i\sum(\omega_i t_i - \vec{k}_i \cdot \vec{x}_i)}, \qquad (1.1)$$

where $Z$ is the partition function and we use the shorthand notation $\delta_\omega \equiv (2\pi)\delta(\sum \omega_i)$, and $\delta_{\vec{k}} \equiv (2\pi)^{d-1}\delta(\sum \vec{k}_i)$ to indicate the delta functions that appear because the position-space correlator on the right is invariant under overall spacetime translations.

The correlator above is a Wightman correlator which means that we just evaluate the quantum expectation value of the product of operators shown and do not impose time-ordering. We now consider the limit of this correlator where $|\vec{k}_i| \to \infty$ but $\omega_i = $ fixed.

About the vacuum, this limit would just yield zero since the spectrum condition, $H \geq |\vec{P}|$, in relativistic quantum field theories tells us that we cannot have excitations with energy smaller than momentum. However, in a thermal state such excitations can exist. For example, the operator $\mathcal{O}$ could create a particle with large momentum $\vec{k}$ and simultaneously destroy a particle from the thermal background with the opposite momentum. This would change the momentum of the state by a large amount but the energy by only a small amount.

In this paper, we show that the correlators (1.1) are constrained by a beautiful geometric bound. We prove, using analytic properties of correlation functions that hold in any relativistic quantum field theory, that

$$\lim_{\substack{|\vec{k}_i|\to\infty \\ \omega_i=\text{const}}} \frac{1}{R(\vec{k}_i)} \log \left| \mathcal{W}(\omega_i, \vec{k}_i) \right| \leq -\beta, \tag{1.2}$$

where $R(\vec{k}_i)$ is the radius of the smallest sphere that encloses the non-planar polygon formed by placing the momenta $\vec{k}_i$ tip to tip. Less formally, our bound states that in this limit, $\mathcal{W}(\omega_i, \vec{k}_i)$ must die off *at least as fast as* $e^{-\beta R(\vec{k}_i)}$. Two examples of this geometric radius are shown in Figure 1; these are relevant for a three-point and a four-point function respectively.

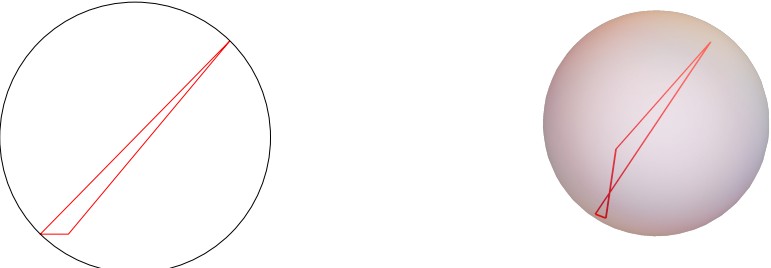

**Figure 1**: *The bounding sphere for a three-point function (left) and a four-point function (right). The red polygon in both cases shows the spatial momenta.*

Our attention was first drawn to this bound because operators whose momentum is much larger than their frequency appear in the map between bulk and boundary operators in the AdS/CFT correspondence [1, 2, 3]. The reconstruction of the field operator at a local bulk point at finite temperature necessarily makes reference to such operators. Such operators also appear if one attempts to reconstruct a causal wedge in the bulk from a boundary causal diamond [4]. This led [4] to analytically continue their bulk-boundary smearing function and has led to some claims that the bulk-boundary map is ill-defined or discontinuous in black-hole backgrounds [5, 6].

This issue was examined in [7] and in [8], it was pointed out that the map was still well-defined since it should be properly thought of as a *distribution* that acts on operators whose natural "size" at large spacelike momenta is small. (This idea was subsequently also elaborated in [9].) For the purpose of the bulk-boundary map at leading order, the only correlator that is relevant is the two-point function and so [7] proved the bound (1.2) for two-point functions. In this paper, we present a generalization of this bound for arbitrary point functions, and also examine its behaviour in various theories.

Correlators of the form (1.1) can also be examined in perturbative weakly-coupled theories. This can be done, as we show, by using thermal field theory techniques and considering the resultant Wick contractions or by setting up a Feynman-diagram formalism as we review in section 3. If we take the operators $\mathcal{O}(t_i, \vec{x}_i)$ to be the elementary fields themselves, then this bound is *not saturated* by the leading tree-level interaction. However, we show in section 3 that once one goes to high-enough loop order, then the perturbative expansion always contains a term that saturates (1.2). For example in a $\phi^3$ scalar theory, the bound for a two-point function is saturated by a one-loop diagram. For a three-point function, and arbitrary external kinematics, the bound is saturated by a two-loop diagram. For some special kinematic configurations, the bound can also be saturated by a one-loop triangle diagram.

This analysis might have led one to believe that holographic theories, which are strongly coupled, would always saturate the bound. However, in our study of holographic large-N theories in section 4, we find a surprise. We consider holographic correlators of operators dual to propagating fields in anti-de Sitter space. Although in $d = 2$, the leading order holographic two-point function in a black-hole background saturates the bound, for higher $d$, the two-point function remains *below* the bound. While the bound would suggest that the two-point function could be as large as $e^{-\frac{\beta |\vec{k}|}{2}}$ it turns out that in $d$-dimensions it is only as large as $e^{-\alpha \frac{\beta |\vec{k}|}{2}}$ where $\alpha = \frac{d\,\Gamma[1+\frac{1}{d}]}{\sqrt{\pi}\Gamma[\frac{1}{2}+\frac{1}{d}]}$. For example, for $d = 4$, $\alpha = 1.67$. This factor was noticed earlier in [10, 5], although no connection to the bound (1.2) was made.

While black-holes are sometimes believed to be "hypercompetitive" [11], this provides an example where the characteristic feature of a strongly coupled holographic theory in a state with a bulk horizon is the *under saturation* of a bound that is saturated by other theories. We do not have an intuitive understanding of this behaviour, and we believe that this is an important and striking feature that deserves further attention.

We also initiate the analysis of $n$-point Wightman functions in large-N holographic theories with gravitational duals. Here, we encounter additional puzzles. Due to the presence of the horizon, the bulk quantum field theory in a black-hole background allows for elementary excitations with arbitrarily small energy but finite momentum. Therefore, its states do not obey the "spectrum condition" — which requires all excited state in a relativistic quantum field theory to have energy larger than the magnitude of their momenta. In fact, as a consequence, if we consider *bulk* correlators at finite values of the radial distance, they violate our bound. On the other hand, boundary correlators can also be defined in the dual field theory and therefore they must obey our bound. However, it is unclear if the bound is obeyed only *nonperturbatively* for $|\vec{k}_i| = \mathrm{O}(N)$ or if it is also obeyed for $|\vec{k}_i| \gg 1$ but $|\vec{k}_i| \ll N$.

This question can be phrased as a question about the analytic properties of holographic correlators defined through Witten diagrams. We analyze these properties by considering when the singularities of bulk to boundary propagators and bulk to bulk propagators can lead to a singularity in the holographic correlator. Through an analysis of geodesics in a complexified black-hole background, we are able to show that at least tree-level contact Witten diagrams obey the bound (1.2). For some cases, such as a BTZ black hole, this

suggests that tree-level exchange diagrams also obey (1.2). However, in general dimensions, we are unable to show that exchange diagrams obey (1.2) and we leave this as an important open question that deserves further attention.

We believe that our bound is interesting and may have other applications. For example, the consideration of correlators with large spacelike momenta can serve as a diagnostic of whether the state is thermal or not. This diagnostic was used in in [12], where it was shown that fuzzball solutions do not saturate the bound even for $d = 2$, suggesting that the known fuzzball solutions are not good representatives of microstates of black-holes and are not dual to the boundary thermal state. The behaviour of correlators of operators with large spacelike momenta may also serve as a diagnostic of when a theory is holographic, and when the bulk has a horizon, just like the chaos bound [13]. This diagnostic was also examined in [14].

A brief overview of this paper is as follows. In section 2, we prove the bound for general relativistic quantum field theories. In section 3, we consider thermal Wightman correlators in perturbative quantum field theories and show that at sufficiently high-loop order we expect such theories to saturate the bound. In 4 we turn to holographic theories. Here, we first analyze holographic two-point functions, and then initiate a study of higher-point functions. The appendices contain a review of the formalism used to compute thermal Wightman functions, and some further details of the holographic analysis.

## 2   Proof of the bound

In this section, we will prove the bound (1.2). We consider a Wightman correlator of the form (1.1) and make the following assumptions

1 The correlator (1.1) is well defined at all values of the temperature.

2 The underlying theory obeys the spectrum condition, so that states which are simultaneous eigenstates of the energy and momentum, with eigenvalues $E$ and $\vec{P}$, satisfy $E \geq |\vec{P}|$.

Due to energy momentum conservation, note that the correlation function (1.1) only has support on the submanifold where $\sum \omega_i = \sum \vec{k}_i = 0$. In particular, this means that the momenta $\vec{k}_i$ generically form a non-planar polygon in $(d - 1)$-dimensions. We now wish to prove the bound (1.2) where $R$ is the radius of the smallest $(d - 1)$-sphere that encloses this polygon.

We will establish the proof in three steps. In the first step we will start with thermal correlators in coordinate space and we will argue that the correlators can be analytically continued to a particular domain of complexified coordinates. In the second step we will show how the analyticity domain of the correlators in coordinate space implies bounds for the momentum space correlators at large spacelike momenta. Finally in the third step, we will show that the optimal such bound is related to a simple geometric extremization problem, whose solution we present. This leads to the bound (1.2) for correlators in momentum space.

## 2.1 First part: on the analyticity domain of position-space thermal correlators

We start with finite temperature, real-time correlators in coordinate space

$$\mathcal{W}(x_i) \equiv Z^{-1}\text{Tr}[e^{-\beta H}\mathcal{O}(x_1)\ldots\mathcal{O}(x_n)]$$

These are Wightman correlators, so there is no time-ordering. For notational convenience we take all operators to be the same; the generalization to different operators is obvious. We use the notation $x$ to denote $d$-vectors and $x^0, \vec{x}$ to denote the timelike and spacelike components respectively. We wish to examine the domain of analyticity of these correlators. Our discussion closely follows [15], and a more detailed discussion is available in [16]. The domain of analyticity for thermal correlators was discussed in [17].

Using translational invariance we can parameterize this correlator as

$$\hat{\mathcal{W}}(\xi_i) \equiv Z^{-1}\text{Tr}[e^{-\beta H}\mathcal{O}(0)\mathcal{O}(\xi_1)\mathcal{O}(\xi_1 + \xi_2)\ldots\mathcal{O}(\xi_1 + \ldots + \xi_{n-1})], \tag{2.1}$$

where $\xi_1 \equiv x_2 - x_1, \xi_2 \equiv x_3 - x_2$ etc. We introduce the timelike vector $e \equiv (1, 0, \ldots, 0)$ and using the cyclicity of the trace we write this correlator as

$$\hat{\mathcal{W}}(\xi_i) = Z^{-1}\text{Tr}[\mathcal{O}(0)e^{-iP\cdot\xi_1}\mathcal{O}(0)e^{-iP\cdot\xi_2}\mathcal{O}(0)\ldots e^{-iP\cdot\xi_{n-1}}\mathcal{O}(0)e^{P\cdot[\beta e + i(\xi_1 + \ldots + \xi_{n-1})]}], \tag{2.2}$$

where $P \equiv (H, \vec{P})$ are the operators for space-time translations and the inner-product between d-vectors is taken using the $(-, + \ldots +)$ metric. We insert complete sets of states between these operators, which leads to an expansion of the form

$$\hat{\mathcal{W}}(\xi_i) = Z^{-1} \sum_{i_1,\ldots,i_n} \mathcal{O}_{i_n i_1} e^{-iP_{i_1}\cdot\xi_1}\mathcal{O}_{i_1 i_2} e^{-iP_{i_2}\cdot\xi_2}\ldots\mathcal{O}_{i_{n-1} i_n} e^{P_{i_n}\cdot[\beta e + i(\xi_1 + \ldots + \xi_{n-1})]}, \tag{2.3}$$

where we defined the matrix elements $\mathcal{O}_{ij} \equiv \langle i|\mathcal{O}(0)|j\rangle$ on eigenstates of the energy-momentum $P$. Now we analytically continue the coordinates $\xi_i$ as $\xi_i \to \xi_i + i\eta_i$. Under this analytic continuation in (2.3) we get factors of the form $e^{P_i\cdot\eta_i}$ in-between the various operators. Using the spectrum condition $H \geq |\vec{P}|$, the factors $e^{P_i\cdot\eta_i}$ will improve the convergence of the sum over $i_1, \ldots, i_{n-1}$ provided that

$$\eta_i \in V^+, \tag{2.4}$$

where $V^+$ denotes the future timelike cone.

On the other hand, the convergence of the sum over $i_n$, corresponding to the overall trace in (2.1), is improved provided that the last factor all the way to the right in (2.3) is suppressed. After analytic continuation that term gives a factor of $e^{P_{i_n}\cdot(\beta e - (\eta_1 + \ldots \eta_{n-1}))}$. From the spectrum condition this improves the convergence provided that

$$\beta e - (\eta_1 + \ldots + \eta_{n-1}) \in V^+. \tag{2.5}$$

Let us call $\mathcal{F}$ the domain of the coordinates $\{\xi_i + i\eta_i\}$ defined by simultaneously imposing equations (2.4) and (2.5) for the imaginary parts $\eta_i$, i.e.

$$\mathcal{F} \equiv \{\{\xi_i + i\eta_i\} \in \mathbb{C}^{4n} : \eta_i \in V^+ \quad \text{and} \quad \beta e - (\eta_1 + \ldots + \eta_{n-1}) \in V^+\}$$

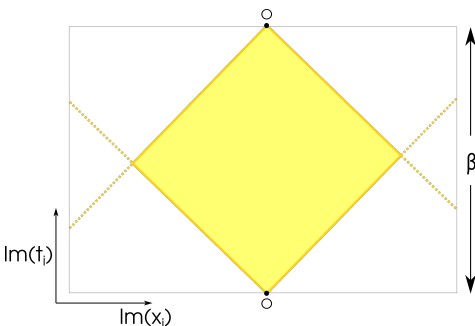

**Figure 2**: *The domain of analyticity $\mathcal{F}$: if the first point is at $O$, then the correlator is analytic if the imaginary coordinates of successive points are chosen from any causal trajectory that lies within the shaded region.*

Notice that $\mathcal{F}$ must be least only *part* of the domain of analyticity of the correlator $\hat{\mathcal{W}}$. This is because by assumption 1 above, the sum over $i_1, \ldots, i_n$ is convergent even for an infinitesimal value of $\beta$, and in the domain, $\mathcal{F}$, the analytic continuation only improves the convergence of the sum. Note that it may be possible to further analytically continue into a larger domain. But, for the purpose of our proof we only need that the correlators $\hat{\mathcal{W}}$ in (2.1) can be analytically continued at least in $\mathcal{F}$ without encountering any singularities. The region $\mathcal{F}$ is shown in Figure 2

Using the notation $\eta_i = (\eta_i^0, \vec{\eta}_i)$, equation (2.4) is equivalent to

$$\eta_i^0 \geq 0 \qquad \text{and} \qquad \eta_i^0 \geq |\vec{\eta}_i|$$

and (2.5) equivalent to

$$\beta - (\eta_1^0 + \ldots + \eta_{n-1}^0) \geq 0$$

and

$$\beta - (\eta_1^0 + \ldots + \eta_{n-1}^0) \geq |\vec{\eta}_1 + \ldots + \vec{\eta}_{n-1}|$$

Using these conditions, we find that any point in the domain $\mathcal{F}$ defined by equations (2.4),(2.5) obeys

$$|\vec{\eta}_1| + \ldots + |\vec{\eta}_{n-1}| + |\vec{\eta}_1 + \ldots + \vec{\eta}_{n-1}| \leq \beta. \tag{2.6}$$

Moreover, it is easy to see that for any choice of the spatial vectors $\vec{\eta}_i$ which satisfies (2.6), we can select their time-components $\eta_i^0$ such that we are inside the domain of (2.4),(2.5). In addition, this choice can be made so that the point $\{\eta^0, \vec{\eta}_i\}$ can be continuously connected to the real-plane $\eta_i = 0$ by a path inside $\mathcal{F}$.

Hence we have established that the domain $\mathcal{F}$ contains points which completely cover the domain in the *spatial coordinates* $\vec{\eta}_i$ defined by equation (2.6).

Now, we remember that $\vec{\eta}_i$ are defined as the spatial part of the imaginary part of the complexified *difference vectors* $\xi_i + i\eta_i$. Let us express the domain of analyticity in terms

of the complexification of the original coordinates $x_i \to x_i + iy_i$. We we have $\vec{\eta}_1 = \vec{y}_2 - \vec{y}_1$ $\vec{\eta}_2 = \vec{y}_3 - \vec{y}_2 \ldots \vec{\eta}_{n-1} = \vec{y}_n - \vec{y}_{n-1}$ Then (2.6) can be written as $|\vec{y}_1 - \vec{y}_2| + \ldots + |\vec{y}_n - \vec{y}_1| \le \beta$.

So finally we reach the following conclusion: the thermal correlator

$$\mathcal{W}(x_i) \equiv Z^{-1} \mathrm{Tr}[e^{-\beta H} \mathcal{O}(x_1) \ldots \mathcal{O}(x_n)]$$

can be analytically continued to complex space-time coordinates $x_i + iy_i$ in (at least) a domain which contains points whose spatial imaginary parts can have any possible value obeying

$$|\vec{y}_1 - \vec{y}_2| + \ldots + |\vec{y}_n - \vec{y}_1| \le \beta. \tag{2.7}$$

Moreover these points are continuously connected to the real-domain $y_i = 0$ by a path in the domain of analyticity.

## 2.2 Second part: decay of spacelike correlators

Now we will explain how the analyticity of position-space correlators discussed above is related to the decay of momentum space correlators at large spacelike momentum.

First we consider "mixed" correlators, where we Fourier transform in time but not in space

$$\mathcal{W}(\omega_i, \vec{x}_i)\delta_\omega \equiv \int \prod_i dt_i \, \mathcal{W}(t_i, \vec{x}_i) e^{i\sum \omega_i t_i}$$

Now we want to analytically continue in $\vec{x}_i$. Notice that we can not just analytically continue the integrand on the RHS in $\vec{x}_i$ alone, as this would — in general— take us out of the domain of analyticity of the correlator. However, we can analytically continue the integrand in $\vec{x}_i$, provided that we shift, at the same time, the time arguments $t_i$ in the complex plane (i.e. we shift the contours of integration by giving nonzero values to $\mathrm{Im}[t_i]$). While doing this we need to make sure that $\mathrm{Im}(t_i, \vec{x}_i)$ stays within the domain of analyticity determined at the end of the previous subsection. Notice that under this analytic continuation the factor $e^{i\omega_1 t_1 + \ldots + i\omega_n t_n}$ gives at most a growing exponential, but can not introduce any singularities.

From this follows that the correlator

$$\mathcal{W}(\omega_i, \vec{x}_i)$$

can be analytically continued as $\vec{x}_i \to \vec{x}_i + i\vec{y}_i$, in (at least) the domain determined by

$$|\vec{y}_1 - \vec{y}_2| + \ldots + |\vec{y}_n - \vec{y}_1| \le \beta. \tag{2.8}$$

Finally we consider the fully Fourier transformed Wightman correlators in frequency momentum space

$$\mathcal{W}(\omega_i, \vec{x}_i) = \int \prod_i \frac{d^{d-1}\vec{k}_i}{(2\pi)^{d-1}} \, \mathcal{W}(\omega_i, \vec{k}_i)\delta_{\vec{k}} e^{i\sum \vec{k}_i \vec{x}_i}$$

We argued above that the LHS is analytic in the domain (2.8) upon $\vec{x}_i \rightarrow \vec{x}_i + i\vec{y}_i$. Under this analytic continuation, on the RHS we get the expression

$$\int \prod_i \frac{d\vec{k}_i}{(2\pi)^{d-1}} \, \mathcal{W}(\omega_i, \vec{k}_i) \delta_{\vec{k}} e^{\sum \vec{k}_i \cdot \vec{y}_i}$$

In general the last factor grows exponentially. In order for the integral to be convergent it must be that $\mathcal{W}(\omega_i, \vec{k}_i)$ decays sufficiently fast at large $\vec{k}_i$, so as to suppress the growth of the last term. This is the origin of the bound (1.2).

The strictest possible bound on $\mathcal{W}$ from these considerations will come from maximizing the expression

$$I \equiv |\sum \vec{k}_i \cdot \vec{y}_i|, \tag{2.9}$$

where

$$|\vec{y}_1 - \vec{y}_2| + \ldots + |\vec{y}_n - \vec{y}_1| \leq \beta, \tag{2.10}$$

and also

$$\sum \vec{k}_i = 0. \tag{2.11}$$

This last condition comes from the fact that the momentum space correlator has support only on momenta obeying this. Let us call $I_{\max}$ the maximum of $I$ defined by (2.9), where we vary the vectors $\{\vec{y}_i\}$ over all possible values, subject to the constraints (2.10). The momentum vectors $\vec{k}_i$ must obey (2.11). Then we find the optimal bound, $\mathcal{W} \sim e^{-I_{\max}}$, or more precisely,

$$\lim_{\substack{|\vec{k}_i| \rightarrow \infty \\ \omega_i = \text{const}}} \frac{1}{I_{\max}} \log(|\mathcal{W}|) \leq -1. \tag{2.12}$$

Determining $I_{\max}$ corresponds to a simple geometric problem that we now consider.

## 2.3 Third part: an extremization problem

Above we found that the momentum-space correlator will have to decay like (2.12), where

$$I_{\max} = \{\max| \sum \vec{k}_i \vec{y}_i| : \vec{y}_i \in \mathbb{R}^{d-1}, |\vec{y}_1 - \vec{y}_2| + \ldots + |\vec{y}_n - \vec{y}_1| \leq \beta\}, \tag{2.13}$$

with $\sum \vec{k}_i = 0$.

It is easy to see that this extremization problem can equivalently be formulated slightly differently by redefining both the $\vec{k}_i$ and the $\vec{y}_i$ variables. First we introduce $n$ new vectors $\vec{P}_i$ such that $\vec{k}_1 = \vec{P}_2 - \vec{P}_1$, $\vec{k}_2 = \vec{P}_3 - \vec{P}_2$, ..., $\vec{k}_{n-1} = \vec{P}_n - \vec{P}_{n-1}$, $\vec{k}_n = \vec{P}_1 - \vec{P}_n$. This does not uniquely fix the $\vec{P}_i$, as we can add to then an overall "center of mass" shift. However, this ambiguity will drop out in what follows. Also, notice that if we parameterize the $\vec{k}_i$'s in terms of $\vec{P}_i$'s as above, then the condition $\sum \vec{k}_i = 0$ is automatic. We also redefine the $\vec{y}$ variables by introducing $\vec{a}_1 \equiv \vec{y}_n - \vec{y}_1$, $\vec{a}_2 \equiv \vec{y}_1 - \vec{y}_2, \ldots, \vec{a}_n \equiv \vec{y}_{n-1} - \vec{y}_n$, where now we automatically have $\sum \vec{a}_i = 0$.

Now, it is a matter of simple algebra to check that the quantity $I = |\sum \vec{k}_i \vec{y}_i|$ that we wanted to extremize takes the form

$$I = \sum |\vec{P}_i \vec{a}_i|, \tag{2.14}$$

where we extremize over $\vec{a}_i$ subject to the condition that $\sum \vec{a}_i = 0$ and the condition (2.10) which becomes

$$\sum |\vec{a}_i| \leq \beta$$

It is now obvious that since $\sum \vec{a}_i = 0$ the overall center of mass shift ambiguity of the $\vec{P}_i$'s that we mentioned earlier has no relevance for the extremization problem.

To summarize, we have shown that the original problem (2.13) is equivalent to the extremization problem

$$I_{\max} = \{\max |\sum \vec{P}_i \vec{a}_i| : \vec{a}_i \in \mathbb{R}^{d-1}, \sum \vec{a}_i = 0, \sum |\vec{a}_i| \leq \beta\}. \tag{2.15}$$

**The solution**:

We now present the solution to the extremization problem (2.15). Consider $n$ points $\vec{P}_i$ in $\mathbb{R}^{d-1}$. We define the "minimal enclosing sphere" in the obvious way. We assume that this sphere has center $\vec{C}$ and radius $R$. The radius $R$ clearly does not depend on the overall center of mass position of the points $\vec{P}_i$.

Consider all possible $n$-tuples of vectors $\vec{a}_i \in \mathbb{R}^{d-1}$, with the properties required in (2.15), i.e. that

$$\sum \vec{a}_i = 0; \qquad \sum |\vec{a}_i| \leq \beta. \tag{2.16}$$

We want to maximize

$$I \equiv |\sum_i \vec{P}_i \vec{a}_i|.$$

We will prove that

$$I_{\max} = \beta R. \tag{2.17}$$

. We first notice that for any $\vec{a}_i$'s obeying (2.16) we have the inequality $I \leq \beta R$ since

$$I \equiv |\sum_i \vec{P}_i \vec{a}_i| = |\sum_i (\vec{P}_i - \vec{C})\vec{a}_i| \leq R(\sum |\vec{a}_i|) \leq \beta R, \tag{2.18}$$

where in the second equality we used the fact that since $\sum \vec{a}_i = 0$ we can shift the overall center of mass of the points by $\vec{C}$ without changing the value of $I$. In the third equality we used $|\vec{P}_i - \vec{C}| \leq R$.

We have shown that for any $\vec{a}_i$ obeying (2.16) we have $I \leq \beta R$. Will now identify a particular choice of $\vec{a}_i$ which saturate the inequality (2.18). This will prove our claim that $I_{\max} = \beta R$.

First we will make use of a basic geometric result: the center $\vec{C}$ of the minimal enclosing sphere of $n$ points $\vec{P}_i$ in $\mathbb{R}^{d-1}$ is in the convex hull of the points $\vec{P}_i$. This means that we can find $n$ real numbers $\lambda_i$ obeying $\lambda_i \geq 0$, $\sum \lambda_i = 1$ such that $\vec{C} = \sum \lambda_i \vec{P}_i$. The proof

of this result is simple: if $\vec{C}$ is not in the convex hull of $\vec{P}_i$ then the *hyperplane separation theorem* says that there is a hyperplane separating $\vec{C}$ from all $\vec{P}_i$. If we move $\vec{C}$ towards this hyperplane we reduce the distance from all points $\vec{P}_i$, contradicting the statement that $\vec{C}$ was the center of the minimal enclosing sphere.

We will now consider a slight refinement of the aforementioned result. For a particular choice of the set of points $\vec{P}_i$ we consider the minimal bounding sphere. Some of the points will be exactly on the sphere, while the remaining will be inside. We concentrate on the $m$ points exactly on the sphere, let us call them *extremal points*. We select the index $i$ labeling the points, so that the extremal points are the first $m$ points $\vec{P}_i, i = 1, \ldots m$, The remaining points which are in the interior of the sphere are labeled as $\vec{P}_i, i = m + 1, \ldots, n$. Notice that it may be that $m = n$. We consider the minimal bounding sphere of the extremal points alone (i.e. simply ignoring the interior points). It should be obvious that the sphere will be exactly the same as before. Applying the previous theorem to the set of extremal points, we conclude that the center $\vec{C}$ of the minimal enclosing sphere is also in the convex hull of the *extremal* points alone. This means that we can write

$$\vec{C} = \sum_{i=1}^{m} \lambda \vec{P}_i, \qquad \lambda_i \geq 0 \quad \text{and} \quad \sum \lambda_i = 1. \tag{2.19}$$

Returning to the extremization problem (2.15), we then consider the following choice of the vectors $\vec{a}_i$

$$\vec{a}_i = \frac{\beta}{R} \lambda_i (\vec{P}_i - \vec{C}) \qquad i = 1, ..m \tag{2.20}$$

and

$$\vec{a}_i = 0, \qquad i = m + 1, \ldots, n. \tag{2.21}$$

It is easy to check, using (2.19) , that the choice of $\vec{a}_i$ given by (2.20) and (2.21) is consistent with conditions (2.16). For this choice we find

$$I = \beta R$$

which saturates the inequality (2.18). Hence we have shown that the solution to the geometric problem is given by (2.17).

From this, using (2.12) and the fact that the minimal size sphere $R$ enclosing the vectors $\vec{P}_i$ has the same radius as the minimal sphere enclosing the polygon of the difference vectors $\vec{k}_i$ placed tip-to-tip, we find the claimed bound (1.2).

## 3    Weakly coupled theories

In this section, we consider the behaviour of thermal Wightman functions in weakly coupled perturbative quantum field theories in the limit where we take the momenta of the insertions to be large and spacelike. For simplicity, we will consider a scalar field, $\phi$, of mass $m$, in a thermal bath at inverse temperature $\beta$ in flat space. However, our analysis can be

easily generalized to weakly coupled gauge theories with a Gauss law constraint by using the techniques of [18].

We are interested in an interaction Hamiltonian that is polynomial in the fields

$$H_I = \sum_n a_n \int \phi_I^n(t, x_i) d^d x_i, \tag{3.1}$$

where $\phi_I$ are the interaction picture operators and $a_n$ the coupling constants. We will present two approaches to analyzing Wightman functions of the field $\phi$ in perturbation theory for the couplings (3.1)— using a straightforward canonical formalism and thinking about (thermal) Wick contractions, or an equivalent set of diagrammatic rules derived using the Schwinger-Keldysh formalism. We explain these in turn.

**Canonical formalism** We consider the Fourier transformed Wightman functions of elementary fields, which are the same as (1.1) except that we focus on the case where the operators, $\mathcal{O}$, are the elementary fields themselves.

$$\mathcal{W}(\omega_i, \vec{k}_i)\delta_\omega \delta_{\vec{k}} \equiv \int \prod_i dt_i d\vec{x}_i \, \frac{1}{Z} \mathrm{Tr}\left[ e^{-\beta H} \phi(t_1, \vec{x}_1) \dots \phi(t_n, \vec{x}_n) \right] e^{i \sum_i \omega_i t_i - \vec{k}_i \cdot \vec{x}_i}, \tag{3.2}$$

where $\phi(t_i, x_i)$ are Heisenberg picture operators and $Z = \mathrm{Tr}(e^{-\beta H})$ is the partition function.

Perturbative Wightman functions can be computed in such theories using the formalism explained in Appendix A. The final result can be expressed in the following form

$$\mathcal{W}(\omega_i, \vec{k}_i)\delta_\omega \delta_{\vec{k}}$$
$$= \sum_{\{s_j\}} \int \prod_{j,l} \frac{d\omega_l^j}{2\pi} \Big( \prod_{j=1}^n 2\pi\delta(\sum_{l=1}^{s_j+1} \omega_l^j - \omega_j) g(\omega_l^j) \Big) \times \frac{1}{Z} \mathrm{Tr}\Big[ (1 + Z_1) e^{-\beta H_0}$$
$$\times [H_I(\omega_1^1), \dots [H_I(\omega_{s_1}^1), \phi_I(\omega_{s_1+1}^1, \vec{k}_1)] \dots] \dots [H_I(\omega_1^n), \dots [H_I(\omega_{s_n}^n), \phi_I(\omega_{s_n+1}^n, \vec{k}_n)] \dots] + Z_2 \Big]. \tag{3.3}$$

Notice that this expectation value is with respect to the thermal density matrix of the unperturbed Hamiltonian $H_0$. The term displayed appears at order $\sum_i s_i$ in perturbation theory, and the leading sum runs over all such terms. In the expression above $g$ is a rational function of the frequencies that is specified in the Appendix but is not important for our asymptotic analysis here.

The terms $Z_1$ and $Z_2$ are subtle terms that arise from infra-red effects in thermal field theory. In the Schwinger-Keldysh formalism, these terms arise from the "vertical part" of the contour as explained in Appendix A. In our calculations below, we will naively assume that

$$(1 + Z_1) = \frac{\mathrm{Tr}(e^{-\beta H})}{\mathrm{Tr}(e^{-\beta H_0})}, \qquad Z_2 = 0. \tag{3.4}$$

In Schwinger-Keldysh language, this corresponds to the assumption that the contribution from the vertical part of the contour decouples. In weakly-coupled field theories, the assumption

(3.4) is believed to be justified provided we use a specific prescription for the two-point function in evaluating Wick contractions [19]. Moreover, we do not believe that the terms $Z_1$ and $Z_2$ will change our conclusions below, which are rather general and not specific to any particular field theory. Nevertheless, (3.4) requires further analysis that we postpone to a later study.

Now, we may expand out the interaction-picture field in terms of creation and annihilation operators

$$\phi_I(t, \vec{x}) = \int \frac{d^{d-1}\vec{k}}{(2\pi)^{d-1}} \frac{1}{\sqrt{2\omega_{\vec{k}}}} \left[ a_{\vec{k}}\, e^{-i\omega_{\vec{k}}t + i\vec{k}\cdot\vec{x}} + a_{\vec{k}}^\dagger e^{i\omega_{\vec{k}}t - i\vec{k}\cdot\vec{x}} \right], \tag{3.5}$$

where $\omega_{\vec{k}} \equiv \sqrt{\vec{k}^2 + m^2}$ and the creation and annihilation operators satisfy

$$[a_{\vec{k}}, a_{\vec{k}'}^\dagger] = (2\pi)^{d-1}\,\delta(\vec{k} - \vec{k}'). \tag{3.6}$$

The thermal correlators of these operators follow from the commutators above using the KMS condition

$$\frac{1}{Z_0}\mathrm{Tr}(e^{-\beta H_0} a_{\vec{k}} a_{\vec{k}'}^\dagger) = \frac{1}{1 - e^{-\beta\omega_{\vec{k}}}}(2\pi)^{d-1}\,\delta(\vec{k} - \vec{k}'),$$
$$\frac{1}{Z_0}\mathrm{Tr}(e^{-\beta H_0} a_{\vec{k}'}^\dagger a_{\vec{k}}) = \frac{e^{-\beta\omega_{\vec{k}}}}{1 - e^{-\beta\omega_{\vec{k}}}}(2\pi)^{d-1}\,\delta(\vec{k} - \vec{k}'), \tag{3.7}$$

where $Z_0 = \mathrm{Tr}(e^{-\beta H_0})$. Note that the interaction Hamiltonian itself can be written as a polynomial in the creation and annihilation operators. Therefore, each commutator of the interaction Hamiltonian with the elementary fields that appears in the expression (3.3) leads to polynomials in the creation and annihilation operators. We denote a general such polynomial comprising *only* products of annihilation operators with $c$-number coefficients as $X(\{\alpha\}, \omega, \vec{k})$. For instance, at quadratic order, an example of such a polynomial with frequency $\omega$ and momentum $\vec{k}$ would be

$$\int_0^\omega d\omega' \int d\vec{k}' a_{\omega',\vec{k}} a_{\omega-\omega',\vec{k}-\vec{k}'}$$

Note that the energy, $\omega$ and momentum $\vec{k}$ of the polynomial is displayed explicitly in our notation. Specifying the frequency and momentum does not uniquely specify the polynomial and we have moved all the rest of the information about the polynomial into the parameter $\{\alpha\}$. This allows us to write

$$g(\omega_n^j)[H_I(\omega_1^j)\dots[H_I(\omega_{s_j}^j), \phi_I(\omega_{s_j+1}^j, \vec{k}_j)]\dots] = \sum_{\alpha,\beta} \int d\omega_1 d\omega_2 d\vec{K}_1 d\vec{K}_2$$
$$\times (2\pi)^d \delta(\omega_1 - \omega_2 - \sum_q \omega_q^j)\delta(\vec{K}_1 - \vec{K}_2 - \vec{k}_j)X(\{\alpha\}, \omega_1, \vec{K}_1)X^\dagger(\{\delta\}, \omega_2, \vec{K}_2) \tag{3.8}$$

We include the rational function $g$ that appears in (3.3) inside the polynomials to lighten the notation. At any order in perturbation theory, the polynomials that appear above can be

systematically computed by using the form of the interaction Hamiltonian (3.1), the expansion (3.5) and the canonical commutators (3.6).

Note that the interaction Hamiltonian itself is integrated over all space, so it does not contribute any momentum, and the momentum on the right hand side comes purely from the insertion of $\phi_I$.

We now need three key facts about these polynomials that appear in the expansion of the Heisenberg picture operators. First, since all the annihilation operators that enter the polynomial are on-shell, this tells us that the integral only has support in the region $\omega > |\vec{k}|$. Second, while the precise correlation functions of these polynomials depend on the specific polynomial under consideration, we note that, generically, as $\omega \to \infty$,

$$
\begin{aligned}
\frac{1}{Z_0}\text{Tr}\left(e^{-\beta H_0}X(\{\alpha\},\omega,k)X^\dagger(\{\delta\},\omega',\vec{k}')\right) &\to \text{O}\,(1)\,\delta(\omega-\omega')\delta(\vec{k}-\vec{k}'), \\
\frac{1}{Z_0}\text{Tr}\left(e^{-\beta H_0}X^\dagger(\{\alpha\},\omega,k)X(\{\delta\},\omega',\vec{k}')\right) &\to \text{O}\left(e^{-\beta\omega}\right)\delta(\omega-\omega')\delta(\vec{k}-\vec{k}').
\end{aligned}
\tag{3.9}
$$

These correlators follow from the elementary thermal correlators (3.7). We have suppressed the dependence on $\{\alpha\}$ and $\{\delta\}$ in the right hand side of the second line although in any concrete calculation this dependence is important. It is possible to choose $\{\alpha\}$ and $\{\delta\}$ so that this coefficient is zero. The correlator is non-zero when the polynomials have the property that their constituent operators can be paired with each other as in (3.7) and, in the equation above, this is understood to be the case. Third, in a general correlation function one might have $n$-point correlators of such polynomials. These correlators can be combined by expanding each polynomial in its constituent creation and annihilation operators, and then using Wicks theorem.

We show below how this data is enough to argue that, at sufficiently high order in perturbation theory, the bound (1.2) is saturated. To lighten the notation we now denote $\frac{1}{Z_0}\text{Tr}(e^{-\beta H_0}O) \equiv \langle O \rangle_\beta$.

**Diagrammatic rules**   As explained in Appendix A, the canonical formalism above can be recast in a set of diagrammatic rules using the Schwinger-Keldysh formalism [20, 21, 22]. The diagrammatic rules for computing thermal Wightman functions are more elaborate than the rules for computing the most-commonly considered time-ordered vacuum correlators.

In the Schwinger-Keldysh formalism, the subtlety corresponding to the factors of (3.4) corresponds to the fact that, as explained in Appendix A, one must carefully take into account the fact that the Schwinger-Keldysh propagators may receive contributions from very early and very late times. This leads to mixed Euclidean-real-time propagators that connect the vertical part of the Schwinger-Keldysh path-integral to the horizontal part. However, it is believed [19, 23] that this effect can be removed by using a specific prescription for the propagator that we adopt below.

This leads to the following simplified Feynman rules for our scalar theory are as follows and that do *not* account for the vertical part of the contour

1. To compute a $n$-point function we consider $\tilde{n}$-copies of the field, where $\tilde{n} = n$ if $n$ is even and $\tilde{n} = n + 1$ if $n$ is odd. We introduce $n$-different interaction vertices, each of which couples fields of type $i$ only to other fields of type $i$. The $i^{\text{th}}$ vertex comes with a sign of $(-1)^{i+1}$.

2. In position space, all interaction vertices are integrated from time $-\infty$ to $\infty$ and over all space. In frequency space, we just impose energy-momentum conservation at each vertex.

3. There are $\tilde{n}^2$-types of propagators that connect fields of type $i$ to fields of type $j$. For the scalar field, these propagators, in frequency space, are given by

$$
D_{ij}(k) = \begin{cases} -\frac{i}{k^2+m^2-i\epsilon} + 2\pi\delta(k^2+m^2)n(|k^0|) & i = j \text{ and } n-i \text{ even,} \\ \frac{i}{k^2+m^2+i\epsilon} + 2\pi\delta(k^2+m^2)n(|k^0|) & i = j \text{ and } n-i \text{ odd,} \\ 2\pi\theta(k^0)\delta(k^2+m^2) + 2\pi\delta(k^2+m^2)n(|k^0|) & i < j, \\ 2\pi\theta(-k^0)\delta(k^2+m^2) + 2\pi\delta(k^2+m^2)n(|k^0|) & i > j. \end{cases} \tag{3.10}
$$

4. The external legs are fields of type $1 \ldots n$.

In the rules above,

$$
n(|k^0|) \equiv \frac{1}{e^{\beta|k^0|}-1}.
$$

Since the reader may find these rules unfamiliar, we give a very explicit example in the case of the two-point function in Table 1 below.

We also note that for a $n$-point correlator, it is possible to extract information about all $n!$ Wightman functions by considering a smaller basis of correlators and using the KMS relations to cleverly obtain information about other correlators [24, 25, 26]. Since our analysis is very simple, we will not utilize these techniques here although we expect that they may be required for concrete calculations of higher-point functions.

## 3.1 Two-point functions

Let us now consider the example of a two-point function. We will analyze this both using the canonical approach, and the diagrammatic approach.

**Canonical analysis** We consider

$$
\mathcal{W}(\omega_1, \vec{k}_1, \omega_2, \vec{k}_2)\delta_\omega\delta_{\vec{k}} = \sum_{\{\alpha\},\{\delta\},\{\alpha\}',\{\delta\}'} \int \prod_{ij} [d\omega_{ij}\, d\vec{K}_{ij}]\, \mathcal{C},
$$
$$
\mathcal{C} = \langle X(\{\alpha\}, \omega_{11}, \vec{K}_{11})X^\dagger(\{\delta\}, \omega_{12}, \vec{K}_{12})X(\{\alpha\}', \omega_{21}, \vec{K}_{21})X^\dagger(\{\delta\}', \omega_{22}, \vec{K}_{22})\rangle_\beta. \tag{3.11}
$$

In the expression above, we have absorbed the delta functions that appear in (3.3) and (3.8) into the measure, which we denote by the square brackets. The correlator itself gives an

overall energy-momentum conserving delta function, which also appears on the left. These delta functions impose energy-momentum conservation that leads to the constraints

$$
\begin{aligned}
\vec{K}_{11} - \vec{K}_{12} &= \vec{k}_1; \quad \omega_{11} - \omega_{12} = \omega_1 \\
\vec{K}_{21} - \vec{K}_{22} &= \vec{k}_2; \quad \omega_{21} - \omega_{22} = \omega_2 \\
\vec{k}_1 + \vec{k}_2 &= 0; \quad \omega_1 + \omega_2 = 0.
\end{aligned}
\tag{3.12}
$$

The correlator can be calculated in terms of Wick contractions. In particular, one term that appears above is just the product of two-point correlators of polynomials

$$
\begin{aligned}
&\langle X(\{\alpha\}, \omega_{11}, \vec{K}_{11}) X^\dagger(\{\delta\}', \omega_{22}, \vec{K}_{22}) \rangle_\beta \langle X^\dagger(\{\delta\}, \omega_{12}, \vec{K}_{12}) X(\{\alpha\}', \omega_{21}, \vec{K}_{21}) \rangle_\beta \\
&\longrightarrow e^{-\beta \omega_{12}} \delta(\omega_{12} - \omega_{21}) \delta(\omega_{11} - \omega_{22}),
\end{aligned}
\tag{3.13}
$$

for large values of $\omega_{ij}$. This is the limit that is relevant since we recall that the polynomials only have support for $\omega_{ij} > |\vec{K}_{ij}|$. In the limit where the mass becomes unimportant, the support of the polynomials actually starts from $\omega_{ij} \approx |\vec{K}_{ij}|$. This implies that the *largest* term in the two-point Wightman correlator above emerges from minimizing $|\vec{K}_{12}|$ subject to all the delta function constraints above. It is clear that this is maximized when $\vec{K}_{12} = \vec{K}_{21} = \frac{-\vec{k}_1}{2}, \vec{K}_{11} = \vec{K}_{22} = \frac{\vec{k}_1}{2}$.

But this means that in the limit under consideration

$$
\mathcal{W}(\omega_1, \vec{k}_1, \omega_2, \vec{k}_2) \longrightarrow e^{-\frac{\beta |\vec{k}_1|}{2}},
\tag{3.14}
$$

precisely consistent with our bound. Note that, such a term appears already at second order in perturbation theory for any $\phi^n$ interaction.

What we have shown here is that there *exists* a term in the perturbative expansion that saturates the bound. The coefficient of this term depends on the precise polynomials that appear above, and the coefficient of this term could vanish. In fact, as we will see in the study of holographic theories for $d > 2$, these theories do *not* saturate the bound at leading order in bulk perturbation theory despite being strongly coupled in the boundary.

**Diagrammatic analysis** We now consider the example of a $\phi^3$ interaction in some more detail. The Feynman rules for the two-point function are given in Table 1.

The reader will immediately see that these rules give rise to multiple diagrams. However, here we just want to show that the perturbative expansion *contains* a term that saturates the bound and not compute the full two-point function. To this end, we consider the diagram shown in figure 3. The corrections from this diagram can extend the support of the two-point Wightman function to off-shell momenta. In fact, due to the absence of terms which mix different fields in the Lagrangian, figure 3 is the simplest diagram which achieves this feat. For simplicity we will study the behaviour of this diagram in the limit $k^0 \to 0$, similar statements can be made for finite $k^0$. We have

$$
\lim_{|\vec{k}| \to \infty} \frac{\lambda^2}{(\vec{k}^2 + m^2)^2} \int \frac{d^d p}{(2\pi)^d} [2\pi \theta(p^0) \delta(p^2 + m^2) + 2\pi \delta(p^2 + m^2) n(|p^0|)]
$$

$$
[2\pi \theta((k-p)^0) \delta((k-p)^2 + m^2) + 2\pi \delta((k-p)^2 + m^2) n(|k - p^0|)]
$$

| Diagram Element | Value |
|---|---|
| | $D_{11}(k)$ |
| | $D_{12}(k)$ |
| | $D_{21}(k)$ |
| | $D_{22}(k)$ |
| | $i\lambda$ |
| | $-i\lambda$ |

**Table 1**: *Feynman rules for a two-point Wightman function in a $\phi^3$ theory. The explicit expressions for $D_{ij}(k)$ are given in (3.10).*

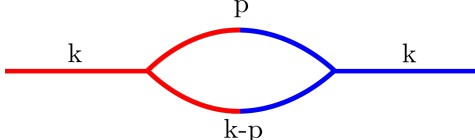

**Figure 3**: *Correction to the 2-point correlator leading to saturation of the bound.*

Even without evaluating this expression exactly, we can estimate its behaviour in the limit of interest as follows. As the external frequency, $k^0$, is negligible, $p$ and $k - p$ must have approximately equal and opposite frequencies. This implies that $\theta(p^0)\theta(k^0 - p^0)$ does not contribute. Moreover, the on-shell condition for the internal propagator implies that $|p^0| = |k^0 - p^0| = |\vec{p}| = |\vec{k} - \vec{p}|$. To estimate the slowest fall off, we want to minimize $|p^0|$. From momentum conservation, vectors $\vec{k}$, $\vec{p}$ and $\vec{k} - \vec{p}$ form a triangle. Imposing $|\vec{p}| = |\vec{k} - \vec{p}|$ would imply that the minimum value of $|p^0|$ is $|\vec{k}|/2$. The Boltzmann factor would become $e^{-\beta|\vec{k}|/2}$. This implies that the diagram is proportional to $e^{-\beta|\vec{k}|/2}$ and therefore, the two-point function saturates our bound at second order in perturbation theory.

This coefficient can be estimated by completing the evaluation of the diagram above and

we find that, in the limit of large-$k$, for $d > 2$, the diagram evaluates to

$$\frac{\lambda^2}{2}\frac{S_{d-3}}{(2\pi)^{d-2}}\frac{|\vec{k}|^{\frac{d}{2}-7}}{\beta^{\frac{d}{2}-1}}\Gamma\left(\frac{d}{2}-1\right)e^{-\beta\frac{|\vec{k}|}{2}}, \tag{3.15}$$

where $S_{d-3}$ is the area of the unit sphere in $d - 3$ dimensions.

## 3.2 Three-point functions

We now move to a consideration of three-point functions and we again perform the analysis in two equivalent ways.

**Canonical analysis**  Just as previously, we now find

$$\mathcal{W}(\omega_1, \vec{k}_1, \omega_2, \vec{k}_2, \omega_3, \vec{k}_3)\delta_\omega\delta_{\vec{k}} = \sum_{\{\alpha\}_q, \{\delta\}_q}\int\prod_{ij}[d\omega_{ij}d\vec{K}_{ij}]\mathcal{C},$$

$$\mathcal{C} = \langle\prod_{q=1}^{3}X(\{\alpha\}_q, \omega_{q1}, \vec{K}_{q1})X^\dagger(\{\delta\}_q, \omega_{q2}, \vec{K}_{q2})\rangle_\beta, \tag{3.16}$$

and we have the constraints

$$\omega_{q1} - \omega_{q2} = \omega_q; \quad \vec{K}_{q1} - \vec{K}_{q2} = \vec{k}_q. \tag{3.17}$$

When the correlator above is expanded using Wick's theorem at finite temperature, we do *not* get only pairwise contractions of the $X$-polynomials, since some annihilation operators in the first polynomial may contract with creation operators from the second $X^\dagger$ whereas some others may contract with creation operators from the third $X^\dagger$. However, of the multiple terms that appear, one particular term that appears in the Wick contraction is

$$\mathcal{C} = \langle X(\{\alpha\}_1, \omega_{11}, \vec{K}_{11})X^\dagger(\{\delta\}_2, \omega_{22}, \vec{K}_{22})\rangle_\beta\langle X^\dagger(\{\delta\}_1, \omega_{12}, \vec{K}_{12})X(\{\alpha\}_3, \omega_{31}, \vec{K}_{31})\rangle_\beta$$

$$\times\langle X(\{\alpha\}_2, \omega_{21}, \vec{K}_{21})X^\dagger(\{\delta\}_3, \omega_{32}, \vec{K}_{32})\rangle_\beta + \dots, \tag{3.18}$$

where ... denote the other possible Wick contractions.

In the displayed term, we find some additional constraints if the term is not to vanish

$$\begin{aligned}\omega_{11} &= \omega_{22}; \quad \vec{K}_{11} = \vec{K}_{22},\\ \omega_{12} &= \omega_{31}; \quad \vec{K}_{12} = \vec{K}_{31},\\ \omega_{21} &= \omega_{32}; \quad \vec{K}_{21} = \vec{K}_{32}.\end{aligned} \tag{3.19}$$

In the limit where the $\omega_1, \omega_2, \omega_3$ are negligible, this just sets all the $\omega_{ij}$ equal to each other.

Now, we note that the displayed term is suppressed by a factor of $e^{-\beta\omega_{11}}$. In the regime where the mass is unimportant, each polynomial only has support in the region where $\omega_{ij} \geq |\vec{K}_{ij}|$ this means that we must have

$$\omega_{11} \geq |\vec{K}_{11}|; \quad \omega_{11} \geq |\vec{K}_{11} - \vec{k}_1|; \quad \omega_{11} \geq |\vec{k}_2 + \vec{K}_{11}|. \tag{3.20}$$

In the expansion of the Wightman function, we must integrate over all values of $\omega_{ij}$ that are allowed. However, the constraints above tell us that the *largest* contribution to the integral comes precisely when $\omega$ takes the smallest value that meets (3.20).This can be achieved by varying $\vec{K}_{11}$ and it is clear that the resultant $\omega_{11}$ is *precisely* the radius of the smallest circle that contains the triangle formed by $\vec{k}_1, \vec{k}_2, \vec{k}_3$.

For any given interaction we can estimate the lowest order in perturbation theory that the term above appears. For instance, consider a $\phi^3$ theory. Then, in general, a non-trivial contraction above first appears at *third order* in perturbation theory, where each of the $X$ polynomials are just single creation and annihilation operators. However, for such operators, the inequalities in (3.20) must actually be equalities since these operators have frequency equal to the norm of their momenta. Now, it is interesting that if the triangle formed by the three momenta $\vec{k}_i$ is acute-angled, then all three-points of the triangle lie on the smallest circle that contains it (the so-called "circumcircle".) Thus, when the triangle formed by $\vec{k}_i$ is acute angled, the minimum value of $\omega$ dictated by (3.20) coincides with the value obtained by saturating all three inequalities. Therefore, for such configurations of momenta, the bound is saturated at third order in perturbation theory for a cubic interaction.

In general, it is *always* possible to keep two points of the triangle on the smallest circle that contains it. This corresponds to taking two out of three contractions in (3.18) to be contractions of single annihilation and creation operators, while taking the third contraction to comprise of polynomials that are at least quadratic in the elementary annihilation and creation operators. The lowest such term appears at *fifth* order in perturbation theory with a $\phi^3$ interaction. On the other hand, if the interaction is $\phi^5$ then such a term appears already at third-order in perturbation theory, and the bound can be saturated for all kinematic configurations at this order.

**Diagrammatic analysis**

The Feynman rules for the three-point function are a natural generalization of the rules above. We are interested in the one-loop diagram shown in Figure 4. This particular loop

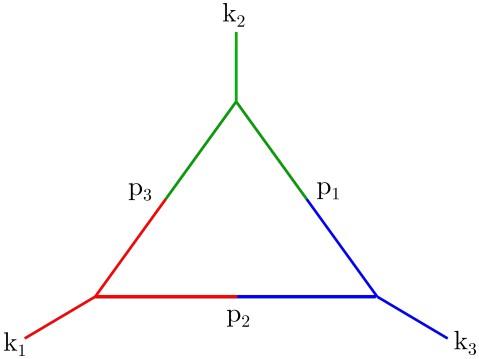

**Figure 4**: *Contribution to the 3-point correlator that saturates the bound for special kinematics*

contribution is given by following integral.

$$\lim_{\vec{k}_i \to \infty} i\lambda^3 \prod \frac{1}{\vec{k}_i^2 + m^2} \int \prod_i \frac{d^d p_i}{(2\pi)^d} \delta(k_1 + p_3 - p_2)\delta(k_2 + p_1 - p_3)\delta(k_3 + p_2 - p_1)$$

$$(2\pi)^3 \delta(p_1^2 + m^2)\delta(p_2^2 + m^2)\delta(p_3^2 + m^2) \left[(\theta(-p_1^0) + n(|p_1^0|)\right] \left[\theta(p_2^0) + n(|p_2^0|)\right] \left[\theta(-p_3^0) + n(|p_3^0|)\right]$$

We are interested in the limit where the masses and external frequencies are negligible. In

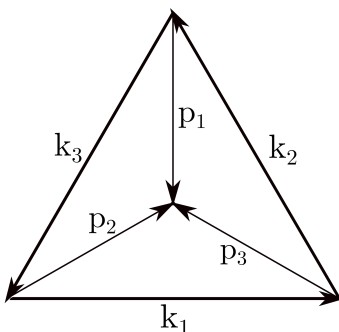

**Figure 5**: *Momentum conservation for diagram 4.*

this limit, the constraints imply that $p_1^0 = p_2^0 = p_3^0$. The on-shell condition, and momentum conservation (see Figure 5) then imposes $|\vec{p}_1| = |\vec{p}_2| = |\vec{p}_3| = R_c$, where $R_c$ denotes circumradius of the triangle with sides $\vec{k}_i$. Due to the presence of both positive and negative signs in theta functions, we are forced to include at least one Boltzmann suppression factor, $e^{-\beta R_c}$. This diagram saturates the bound only for limited kinematic configurations, that is, when the triangle formed by external momenta is acute angled.

In order to saturate the bound in all of kinematic space we require at least one off-shell (but time-like) internal propagator (as we can always keep 2 propagators on-shell and still saturate the bound). This can be achieved by correcting the internal propagator by introducing two additional vertices as shown in Figure 6. Note that no additional large-k suppressions are introduced as we are interested in the regime where the internal propagators are time-like. Hence, for a $\phi^3$ interactions we need $5^{\text{th}}$ order corrections in the coupling to saturate the bound for all of kinematic space.

### 3.3 Higher-point functions

The generalization to higher-point functions is now quite simple.

**Canonical analysis**  For a $n$-point function, we obtain the perturbative series

$$\begin{aligned}
&\mathcal{W}(\omega_i, \vec{k}_i)\delta_\omega \delta_{\vec{k}} \\
&= \sum_{\{\alpha\}_q, \{\delta\}_q} \int \prod_{ij} [d\omega_{ij} d\vec{K}_{ij}] \langle \prod_{q=1}^n X(\{\alpha\}_q, \omega_{q1}, \vec{K}_{q1}) X^\dagger(\{\delta\}_q, \omega_{q2}, \vec{K}_{q2}) \rangle_\beta.
\end{aligned} \tag{3.21}$$

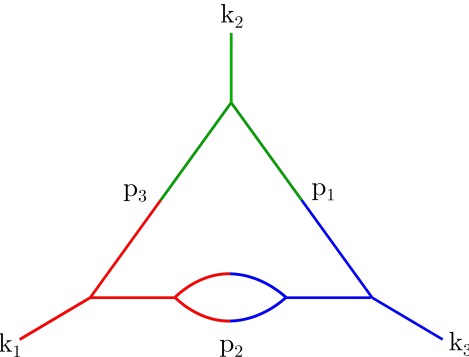

**Figure 6**: *Diagram for the three-point correlator that saturates the bound for all external kinematics*

The integral is subject to the constraints

$$\omega_{q1} - \omega_{q2} = \omega_q; \quad \vec{K}_{q1} - \vec{K}_{q2} = \vec{k}_q. \tag{3.22}$$

Expanding this using Wick's theorem, we now find the following term

$$\begin{aligned}
&\mathcal{W}(\omega_i, \vec{k}_i)\delta_\omega\delta_{\vec{k}} \\
&= \sum_{\{\alpha\}_q, \{\delta\}_q} \int \prod_{ij} [d\omega_{ij} d\vec{K}_{ij}] \left[ \prod_{i=1}^{n-1} \langle X(\{\alpha\}_i, \omega_{i1}, \vec{K}_{i1}) X^\dagger(\{\delta\}_{i+1}, \omega_{i+1,2}, \vec{K}_{i+1,2}) \rangle_\beta \right] \\
&\quad \times \langle X^\dagger(\{\alpha\}_1, \omega_{12}, \vec{K}_{12}) X(\{\delta\}_n, \omega_{n1}, \vec{K}_{n1}) \rangle_\beta + \dots.
\end{aligned} \tag{3.23}$$

These correlators are additionally non-zero when the constraints

$$\omega_{i1} - \omega_{i+1,2} = 0; \quad \vec{K}_{i1} - \vec{K}_{i+1,2} = 0, \tag{3.24}$$

are satisfied. The only term that is exponentially suppressed in the expression above appears on the second line. In the limit where $\omega_i \ll |\vec{k}_i|$, it is clear that the *smallest* value of $\omega_{12}$ that satisfies the constraints is the radius of the smallest sphere that contains the polygon formed by the $\vec{k}_i$ precisely in line with our bound.

As in the discussion of the three-point function, since we can always place at least two points from this polygon on the sphere itself, this implies that two of the contractions in (3.23) (which involve four polynomials) can comprise polynomials of order 1. However the other $(2n-4)$ polynomials must be quadratic or higher. For a $\phi^3$ interaction, this means that such a term first appears at order $3n-4$ in perturbation theory. For a $\phi^5$ interaction on the other hand, such a term appears already at order $n$ in perturbation theory.

**Diagrammatic analysis** It is also simple to see the diagram that contributes the relevant term in the $n$-point function. First, we need a minimum of n-th order correction to allow all the external momenta to be spacelike. Now we need $n-2$ internal momenta to be off-shell.

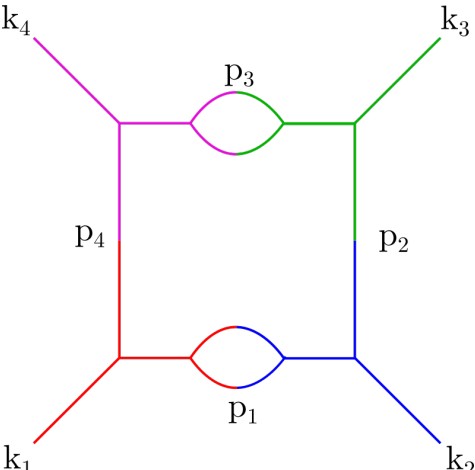

**Figure 7**: *The bound for four-point function is expected to be saturated at eighth order in perturbation theory for arbitrary external kinematics by the diagram above.*

This, in $\phi^3$ theory, would require additional $2(n-2)$ vertices. So an n-point function will saturate the bound at order $3n-4$ in perturbation theory. For instance, Figure 7 shows the diagram that saturates the bound for 4-point function for the $\phi^3$ theory.

If we work with $\phi^5$ or higher interactions, then we can saturate the bound for n-point functions at n-th order in perturbation theory. This is because we can keep all internal momenta off-shell without introducing additional vertices, as shown, for instance, in figure 8.

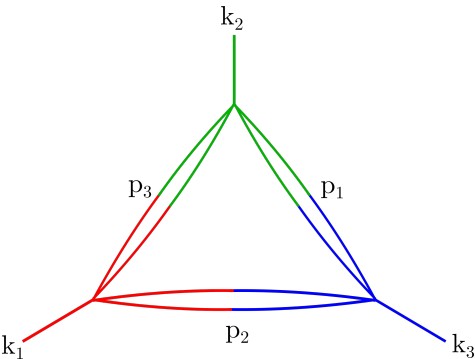

**Figure 8**: *A $\phi^5$ interaction can saturate the bound for n-point functions at $n^{th}$ order in perturbation theory. The diagram above shows the relevant correction to the 3-point function.*

In the case of a $\phi^4$ interaction, odd-point functions vanish. Also, half of the internal momenta can be made off-shell by the same logic as above. An even $n > 2$ point function could saturate the bound at order $n + 2(n - 2 - n/2) = 2n - 4$ in perturbation theory.

In this section, we have considered correlators of elementary fields in weakly interacting theories. However, as the canonical analysis above makes clear, even in a *free theory*, correlators of suitably complicated composite operators saturate the bound.

## 4 Holographic theories

We now consider a large-N field theory with a gravitational holographic dual. In such a theory, the natural low-energy operators are generalized free-fields. So in this section, we analyze the behaviour of correlation functions of generalized free-fields at large spacelike momenta. While our proof of the large-k bound is valid for such theories, there is a subtlety. The bound proved in 2 is strictly valid only for asymptotically large momenta. If we then consider a holographic correlator where the insertions have momenta such that $|\vec{k}| \gg \omega, |\vec{k}| \gg T$ (where $T$ is the temperature), but also $|\vec{k}| \ll N$, is the bound obeyed, or is the bound valid only for $|\vec{k}| \gg N$. In this section, we will make some progress towards understanding this question, but we will not reach a final answer.

We analyze both two-point functions and higher-point functions in a holographic theory at finite temperature. The two-point function can be analyzed by considering the propagation of bulk fields on top of a black-brane background. An interesting aspect of the holographic correlators, so obtained, is that their Fourier transforms have *non-zero* support in the regime of our interest (large spacelike momenta), even when the bulk theory is described by a free-field propagating in the curved background. We will show that, at this level, while the two-point function saturates the bound (1.2) for AdS$_3$, it remains *strictly below* the bound in higher dimensional AdS. We do not understand the reason for this curious behaviour particularly since the analysis of section 3 might have suggested that in a generic strongly coupled theory (where all orders in the perturbative expansion are important) the bound is always saturated.

In the second part of the section, we initiate the study of interactions for holographic correlators. Interactions introduce several dangerous terms in the bulk perturbative expansion that have the potential to violate the bound. This is because the bulk theory does *not* obey the spectrum condition and has excitations with frequency smaller than momentum. Nevertheless, by means of an analysis of the analytic structure of Witten diagrams, we argue that tree-level *contact* Witten diagrams do obey the bound. We make some brief comments about exchange diagrams in 4.2.2.

We should emphasize that at all points in this section, we are only interested in *boundary correlators*. Bulk correlators, where the operators are inserted at some finite value of the radial coordinate manifestly violate our bound. This is because these operators do *not* have well-defined correlators at all temperatures, which is one of the assumptions in our proof. In particular, if we keep the radial position of the operator fixed, and increase the temperature the operator will eventually fall behind the horizon, where it must be described by the state-dependent operators of [8, 27, 28, 29].

## 4.1 Two-point functions

To analyze the behaviour of two-point functions of generalized free-fields, we consider free-fields in AdS, propagating on black brane backgrounds. For simplicity, we will only consider scalar fields. A similar analysis was also performed in [5].

We set the radius of AdS to 1 so that the metric of the black-brane in AdS is given by

$$ds^2 = \frac{1}{z^2}\left[-h(z)dt^2 + \frac{1}{h(z)}dz^2 + d\vec{x}^2\right], \tag{4.1}$$

where

$$h(z) = 1 - \frac{z^d}{z_0^d}.$$

The horizon is at $z = z_0$, the boundary at $z = 0$ and $\vec{x}$ is a $(d-1)$-dimensional vector. The inverse-temperature of this brane is given by

$$\beta = \frac{4\pi z_0}{d}. \tag{4.2}$$

We will consider a massive scalar field in the bulk that satisfies the wave-equation

$$(\Box - m^2)\phi = 0. \tag{4.3}$$

To analyze this wave-equation, it is useful to switch to coordinates defined by[1]

$$\frac{dz_*}{dz} = \frac{1}{zh(z)}. \tag{4.4}$$

The map and inverse-map between $z_*$ and $z$ is given by

$$z_* = -\frac{\log\left(z_0^d - z^d\right)}{d} + \frac{\log\left(z^d\right)}{d}; \quad z = \frac{z_0}{\left(1 + e^{-dz_*}\right)^{\frac{1}{d}}}. \tag{4.5}$$

We make an ansatz of the form $\phi = \chi_{\omega,\vec{k}}(z)e^{i\vec{k}\cdot x - i\omega t}$. Further, it is convenient to substitute $\chi_{\omega,\vec{k}}(z) = z^{\frac{d}{2}}\psi(z)$, where we suppress the dependence of $\psi$ on $\omega, \vec{k}$ to lighten the notation. We now find that $\psi(z)$ obeys the equation

$$\frac{d^2\psi}{dz_*^2} + V\psi = 0, \tag{4.6}$$

with

$$V = z^2\omega^2 + h(z)\left(-\frac{d^2}{4} - m^2 - z^2\vec{k}^2 - \frac{d^2 z^d}{4z_0^d}\right). \tag{4.7}$$

We now consider this equation in the limit that $|\vec{k}| \to \infty$ with $\omega$ fixed.

---

[1]Notice that the coordinate $z_*$ is not the same as the tortoise coordinate.

For this regime of parameters, it is most convenient to solve the equation in the following three regions

$$\text{Region I:}\quad \frac{z_0 - z}{z_0} \ll 1,$$
$$\text{Region II:}\quad h(z) \gg \frac{\omega^2}{|\vec{k}|^2}\quad \text{and}\quad |\vec{k}|^2 z^2 \gg 1, \tag{4.8}$$
$$\text{Region III:}\quad z \ll 1,$$

and then match the solutions in their overlapping regimes of validity. The potential has a turning point but this is included in region I above.

### 4.1.1 Approximate solution in different regions

**Region I —** In this region we find that $z_* \gg 1$ and we can approximate

$$z \approx z_0(1 - \frac{1}{d}e^{-dz_*}). \tag{4.9}$$

We can also approximate the potential as

$$V \approx z^2 \left(\omega^2 - h\vec{k}^2\right) \approx z_0^2 \left(\omega^2 - e^{-dz_*}\vec{k}^2\right). \tag{4.10}$$

This leaves us with the differential equation

$$\ddot{\psi} + z_0^2 \left(\omega^2 - e^{-dz_*}\vec{k}^2\right)\psi = 0,\quad \text{in Region I,} \tag{4.11}$$

This is just the modified Bessel equation (although the order is imaginary) and with $\gamma = 2z_0 d^{-1}$, the solution is

$$\psi = A_I K_{i\gamma\omega}(|\vec{k}|\gamma e^{-dz_*/2}) + B_I \mathcal{R}\left(I_{i\gamma\omega}(|\vec{k}|\gamma e^{-dz_*/2})\right), \tag{4.12}$$

where $\mathcal{R}$ yields the real part of its argument.

**Region II —** In region II, we will use a WKB approximation to solve the equation. We approximate the potential by

$$V \approx -z^2 h|\vec{k}|^2,\quad \text{Region II} \tag{4.13}$$

Then, with,

$$W(z) = \int \sqrt{-V}dz_* = \int \sqrt{-V}\frac{dz}{zh(z)} = |\vec{k}|z\,_2F_1\left(\frac{1}{2}, \frac{1}{d}; 1 + \frac{1}{d}; \left(\frac{z}{z_0}\right)^d\right), \tag{4.14}$$

the solution is given by

$$\psi(z) = \frac{1}{(-V)^{\frac{1}{4}}}\left(A_{II}e^{W(z)} + B_{II}e^{-W(z)}\right). \tag{4.15}$$

**Region III—** In region III, we can neglect the non-linear terms inside $h$ and approximate the potential by

$$V = -|\vec{k}|^2 z^2 - \left(\frac{d}{2}\right)^2 - m^2. \tag{4.16}$$

In this region, we also have

$$z = z_0 e^{z_*}, \quad \text{Region III}, \tag{4.17}$$

The solution to the differential equation is

$$\psi = A_{III} I_\nu(|\vec{k}|z) + B_{III} K_\nu(|\vec{k}|z), \tag{4.18}$$

with $\nu = \sqrt{\left(\frac{d}{2}\right)^2 + m^2}$.

### 4.1.2 Matching

We now match the three solutions, given in (4.12), (4.15), (4.18) to relate the constants above to each other.

Now, as we enter the range where $|\vec{k}|z \gg 1$, but $z \ll 1$, which is the overlap between region III and region II, we find that the solution can be written by considering the asymptotics of both (4.18) and of (4.15). In this region,

$$\psi = \frac{A_{III}}{\sqrt{2\pi|\vec{k}|z}} e^{|\vec{k}|z} + \frac{e^{-|\vec{k}|z}}{\sqrt{2\pi|\vec{k}|z}} \left(\pi B_{III} + i A_{III} e^{i\pi\nu}\right) = \frac{1}{\sqrt{|\vec{k}|z}} \left(A_{II} e^{|\vec{k}|z} + B_{II} e^{-|\vec{k}|z}\right). \tag{4.19}$$

There is a subtlety about whether we should match both the positive and the negative exponential terms in (4.19). However, note that first the leading constants $A_{II}$ and $B_{II}$ that multiply these terms could make them of the same magnitude. Second, we can also imagine continuing the exponential into the imaginary plane so that the exponents become phases, and then match them.

This leads to the relations

$$A_{II} = \frac{A_{III}}{\sqrt{2\pi}}; \quad B_{II} = \frac{\pi B_{III} + i A_{III}}{\sqrt{2\pi}}. \tag{4.20}$$

Now we turn to region I. First, by extending from region II towards region I, we find

$$W(z) \xrightarrow{z \to z_0} \frac{|\vec{k}|\sqrt{\pi}z_0 \Gamma[1 + \frac{1}{d}]}{\Gamma[\frac{1}{2} + \frac{1}{d}]} - 2|\vec{k}|d^{-\frac{1}{2}} z_0^{\frac{1}{2}} \sqrt{z_0 - z}. \tag{4.21}$$

For convenience below we define

$$\kappa = \frac{|\vec{k}|\sqrt{\pi}z_0 \Gamma[1 + \frac{1}{d}]}{\Gamma[\frac{1}{2} + \frac{1}{d}]}. \tag{4.22}$$

So, the WKB solution, as we approach region I becomes

$$\psi = \frac{1}{\sqrt{|\vec{k}|z_0 e^{-dz_*/2}}} \left(A_{II} e^{\kappa - 2|\vec{k}|d^{-\frac{1}{2}} z_0^{\frac{1}{2}} \sqrt{z_0 - z}} + B_{II} e^{-\kappa + 2|\vec{k}|d^{-\frac{1}{2}} z_0^{\frac{1}{2}} \sqrt{z_0 - z}}\right). \tag{4.23}$$

On the other hand, we have a regime where $|\vec{k}|e^{-dz_*} \gg 1$, but nevertheless, $z_* \gg 1$ that overlaps with the regime above and is part of region I. This happens for $1 \ll z_* \ll \ln(|\vec{k}|)$. In this region, the expansion of the Bessel functions is

$$
\begin{aligned}
K_{i\gamma\omega}(\gamma|\vec{k}|e^{-dz_*/2}) &\approx \frac{1}{\sqrt{\frac{4}{d\pi}|\vec{k}|z_0 e^{-dz_*/2}}} e^{-2z_0|\vec{k}|d^{-1}e^{-dz_*/2}} \\
\mathcal{R}\left(I_{i\gamma\omega}(\gamma|\vec{k}|e^{-dz_*/2})\right) &\approx \frac{1}{\sqrt{\frac{4\pi}{d}|\vec{k}|z_0 e^{-dz_*/2}}} e^{2z_0|\vec{k}|d^{-1}e^{-dz_*/2}} .
\end{aligned}
\tag{4.24}
$$

We can match these asymptotics with the asymptotics of region II, by using the fact that in the overlapping region

$$
e^{-\frac{dz_*}{2}} = \sqrt{d(1-\frac{z}{z_0})} = d^{\frac{1}{2}} z_0^{-\frac{1}{2}} \sqrt{z_0 - z}.
\tag{4.25}
$$

Therefore the solution from region I as we approach region II becomes

$$
\psi = A_I \frac{1}{\sqrt{\frac{4}{\pi d}|\vec{k}|z_0 e^{-dz_*/2}}} e^{-2|\vec{k}|z_0^{\frac{1}{2}} d^{-\frac{1}{2}}\sqrt{z_0-z}} + B_I \frac{1}{\sqrt{\frac{4\pi}{d}|\vec{k}|z_0 e^{-dz_*/2}}} e^{2|\vec{k}|z_0^{\frac{1}{2}} d^{-\frac{1}{2}}\sqrt{z_0-z}}.
\tag{4.26}
$$

Now matching (4.26) and (4.23) we see that we need

$$
\begin{aligned}
A_I &= \sqrt{\frac{4}{\pi d}} e^\kappa A_{II}, \\
B_I &= \sqrt{\frac{4\pi}{d}} B_{II} e^{-\kappa}.
\end{aligned}
\tag{4.27}
$$

Combining (4.27) and (4.20) we find that

$$
\begin{aligned}
A_{III} &= \pi A_I e^{-\kappa} \sqrt{\frac{d}{2}}, \\
B_{III} &= \sqrt{\frac{2}{\pi}} (B_{II} - iA_{II}) = \sqrt{\frac{2}{\pi}} \left(e^\kappa B_I \sqrt{\frac{d}{4\pi}} - ie^{-\kappa} A_I \sqrt{\frac{\pi d}{4}}\right).
\end{aligned}
\tag{4.28}
$$

### 4.1.3   Normalization

Finally, we need to normalize the solutions above so that they can be used as a basis for expanding a quantum field. First, near the boundary we note that

$$
\begin{aligned}
\psi &= A_{III} I_\nu(|\vec{k}|z) + B_{III} K_\nu(|\vec{k}|z) \\
&\xrightarrow{z\to 0} (|\vec{k}|z)^\nu \left(B_{III}\Gamma(-\nu)2^{-\nu-1} + A_{III}\frac{2^{-\nu}}{\Gamma(1+\nu)}\right) + (|\vec{k}|z)^{-\nu} 2^{\nu-1}\Gamma(\nu)B_{III}.
\end{aligned}
\tag{4.29}
$$

Since we are looking for normalizable solutions, we set

$$
B_{III} = 0.
\tag{4.30}
$$

This also tells us that $|B_I| \ll |A_I|$ in the large $|\vec{k}|$ limit, and so we can neglect $B_I$ in what follows. Next, in the region near the horizon, where $z_* \gg 1$, we have the expansion

$$K_{i\gamma\omega}(\gamma|\vec{k}|e^{-dz_*/2}) \xrightarrow{z_* \to \infty} -\left(\frac{\pi}{\gamma\omega\sinh(\pi\gamma\omega)}\right)^{1/2} \sin(-d\gamma\omega z_*/2 + \log(\gamma|k|/2)\gamma\omega - \delta). \quad (4.31)$$

where the phase $\delta = \arg\left(\Gamma(1 + i\gamma\omega)\right)$ and we have ignored the expansion of the Bessel "I" function since $B_I$ is negligible.

We can use this to set the normalization of the field as follows. We expand the bulk quantum field as

$$\phi = \int \frac{d\omega d^{d-1}\vec{k}}{(2\pi)^d} \frac{1}{\sqrt{2\omega}} a_{\omega,\vec{k}}\psi(z)e^{-i\omega t}e^{i\vec{k}\cdot\vec{x}}, \quad (4.32)$$

with the creation and annihilation operators normalized so that

$$[a_{\omega,\vec{k}}, a^\dagger_{\omega',\vec{k}'}] = (2\pi)^d \delta(\omega - \omega')\delta(\vec{k} - \vec{k}'). \quad (4.33)$$

The correct normalization of $\psi_{\omega,\vec{k}}(z)$ can then be determined through the canonical commutation relations

$$[\phi(t, z, \vec{x}), g^{tt}\dot{\phi}(t, z', \vec{x}')] = \frac{i}{\sqrt{-g(z_*)}}\delta(z_* - z'_*)\delta(\vec{x} - \vec{x}') \quad (4.34)$$

By examining these commutation relations in the near-horizon region where the wave-function varies exponentially, we find that

$$A_I^2 = \frac{8z_0^d\omega\sinh(\frac{2\pi z_0\omega}{d})}{\pi}$$

### 4.1.4 Two-point functions

The analysis above permits us to calculate the two-point correlation function of the generalized free-field on the boundary, $\mathcal{O}$, that is dual to the bulk field $\phi$ through

$$\langle \mathcal{O}_{\omega,\vec{k}}\mathcal{O}_{\omega',\vec{k}'}\rangle_\beta = \lim_{z \to \infty} z^{-2\Delta}\langle \phi_{\omega,\vec{k}}(z)\phi_{\omega',\vec{k}'}(z)\rangle_\beta. \quad (4.35)$$

Note that this two-point function is sometimes defined with a "wave-function renormalization" factor that we have set to 1. The quantum expectation value on the right hand-side can be computed by using

$$\langle a_{\omega,\vec{k}}a^\dagger_{\omega',\vec{k}'}\rangle = \frac{1}{1 - e^{-\beta\omega}}(2\pi)^d\delta(\omega - \omega')\delta(\vec{k} - \vec{k}');$$
$$\langle a^\dagger_{\omega,\vec{k}}a_{\omega',\vec{k}'}\rangle = \frac{e^{-\beta\omega}}{1 - e^{-\beta\omega}}(2\pi)^d\delta(\omega - \omega')\delta(\vec{k} - \vec{k}'). \quad (4.36)$$

This leads to the result

$$\lim_{|\vec{k}| \to \infty}\langle \mathcal{O}_{\omega,\vec{k}}\mathcal{O}_{\omega',\vec{k}'}\rangle_\beta = 2\pi dz_0^d\cosh(\frac{2\pi z_0\omega}{d})|\vec{k}|^{2\nu}\frac{2^{-2\nu}}{\Gamma(1+\nu)^2}e^{-2\kappa}\delta(\omega + \omega')\delta(\vec{k} + \vec{k}'). \quad (4.37)$$

Apart from some leading constants, the important part of this result for us is that in the large-k limit, the two-point function scales like $e^{\frac{-\alpha\beta|\vec{k}|}{2}}$ where[2]

$$\alpha = \frac{d\Gamma[1 + \frac{1}{d}]}{\sqrt{\pi}\Gamma[\frac{1}{2} + \frac{1}{d}]}. \tag{4.38}$$

While, for $d = 2$, we have $\alpha = 1$ for $d > 2$, we have $\alpha > 1$. In particular, for $d = 3, 4, 5, 6$ we have $\alpha = 1.34, 1.67, 2.00, 2.32$ respectively.

This means that while the bound is saturated in $d = 2$, it is under-saturated for $d > 2$. It would be nice to understand the reason for this phenomenon.

## 4.2 Interactions and higher-point functions

We now examine how the correlators above behave when interactions are included. We will prove that, at tree-level, holographic correlators computed via *contact Witten diagrams obey the bound* (1.2). Our arguments do not immediately show that the bound is saturated, and we postpone a more-complete discussion of exchange diagrams to future work. Interactions in holographic theories at finite temperature have been considered extensively in the literature starting with the work of [10, 30]. We refer the reader to [31, 32, 33, 34, 35] for more details.

Our analysis proceeds as follows. We consider Witten diagrams in the background of the *Euclidean black brane.* The Euclidean black brane metric is given by the continuation of (4.1)

$$ds_E^2 = \frac{1}{z^2}\left[h(z)d\tau^2 + \frac{1}{h(z)}dz^2 + d\vec{x}^2\right], \tag{4.39}$$

with a periodic identification of Euclidean time through $\tau \sim \tau + \beta$. This metric is completely regular and the $\tau$-circle shrinks smoothly to zero at $z = z_0$. For notational consistency we will continue to use the coordinate $t = -i\tau$.

In this section, we assume that the boundary correlator at real time and finite temperature can be computed as follows

1. We integrate all bulk points and bulk to bulk propagators over the Euclidean black-brane geometry.

2. We analytically continue the bulk to boundary propagators to account for complexified positions of the boundary insertions.

This seems to be a natural prescription for computing finite-temperature, real time correlators and avoids some of the difficulties that appear in the Schwinger-Keldysh formalism, which are explained in Appendix B.

For simplicity, we will consider scalar fields dual to operators of dimension $\Delta$. We consider contact interactions in some detail, and then briefly mention exchange interactions.

---

[2]A similar factor appears in [5], although our expression is different. The discrepancy may be due to a typographical error.

### 4.2.1 Contact interactions

Witten diagrams with contact interactions can be computed using the bulk-boundary propagator in this background, $K_\Delta(t_0, \vec{x}_0, t, \vec{x}, \vec{z})$ from a boundary point $(t_0, \vec{x}_0)$ to a bulk point $(t, \vec{x}, z)$ and a typical diagram is evaluated through an integral of the form

$$W(t_i, x_i) = \int \prod_i K_{\Delta_i}(t_i, \vec{x}_i, t, \vec{x}, z) dt d^{d-1}\vec{x} \frac{dz}{z^{d+1}}, \tag{4.40}$$

where the contour of integration is

$$0 \le z \le z_0; \quad \vec{x} \in R^{d-1} \quad 0 \le it \le \beta.$$

The purely Euclidean computation would involve purely imaginary values for the boundary points $t_i$ and purely real values for $x_i$. For such values, the bulk to boundary propagator has *no* singularities. However, here, we will allow the boundary points to be at general complex values of $t_i, \vec{x}_i$, which can be done by analytically continuing the bulk to boundary propagator.

Now, the key point is as follows. As we start with Euclidean boundary points and continue them to complex values, the bulk to boundary propagator in (4.40) may develop singularities. Nevertheless, the *integral* itself can usually still be defined through analytic continuation. The *integral* develops singularities only when the contour of integration gets *pinched* between two or more singularities of the integrand [36].[3]

Although, in general, we cannot find explicit analytic expressions for the bulk-boundary propagator for higher than two boundary dimensions, we can still isolate its singularities. The bulk-boundary propagator is singular whenever the boundary point $(t_i, x_i)$ is connected to the bulk point $(t, x, z)$ by a null geodesic. Since the boundary points are at complex positions, we consider *complexified* geodesics.

**Conditions for the Contour to be Pinched**   We now review the conditions under which the contour of integration may be pinched. Let the equation of the light-cone emanating from a boundary point $(t_i, x_i)$ to a boundary point $(t, x, z)$ be given by $S_i = 0$. Then for the integral (4.40) to be singular, we require the following *necessary condition*. For some $q$ *distinct* values $i_1, i_2 \ldots i_q$, we should have

$$S_{i_1} = S_{i_2} = \ldots S_{i_q} = 0; \quad \sum_{j=1}^q \gamma_j \frac{\partial S_{i_j}}{\partial w} = 0; \tag{4.41}$$

where $\gamma_1, \gamma_2 \ldots \gamma_q$ are arbitrary complex numbers and $w_k$ runs over $t, \vec{x}, z$.

The first condition in (4.41) expresses the fact that the singularities are coincident. The second condition expresses the fact that the normals to the light-cone at the point of coincidence are linearly dependent on each other.

---

[3]This is similar to the method used in [37] to locate singularities in holographic correlators. However, since the bulk background is that of a black brane rather than empty AdS, the analysis here is considerably more involved.

In addition, it is important that the singularities do not approach the contour of integration from the "same" side. Let $\delta \vec{x}_i = x_i - x$ and $\delta t = t_i - t$. Then we require the following condition: *if the vectors $Im\,(\delta \vec{x}_i)$ are on the same side of any $(d-2)$-dimensional hyperplane that runs through the origin, then the singularity is not pinched.* Mathematically, this condition can be expressed by stating

$$\nexists \vec{b} \in R^{d-1}, \quad \text{such that} \quad Im\,(\delta \vec{x}_i) \cdot \vec{b} > 0, \ \forall i. \tag{4.42}$$

The reason for this is that, in such a case, by deforming the contour of integration to give $\vec{x}$ a small imaginary part in the direction perpendicular to this hyperplane, we simultaneously move away from all singularities.

We now prove that in a contact Witten diagram, the contour of integration *cannot* be pinched between the singularities of the bulk-boundary propagators.

**Sketch of Proof** The proof below is somewhat involved, so we provide a brief sketch of the steps involved.

1. First we show that, if the contour of integration lies on a real value of $z$ then singularities of the analytically continued bulk-boundary propagator only occur when the imaginary part of the displacement from the boundary to the bulk point is null or spacelike:

$$[Im(\delta t)]^2 \leq Im(\delta \vec{x}) \cdot Im(\delta \vec{x}). \tag{4.43}$$

2. Simple geometry then shows that if the boundary points are in the domain of analyticity, then (4.42) cannot be met.

**An analysis of complexified geodesics in higher-dimensional black branes**

The null geodesic equations, written in terms of an affine parameter, $\lambda$ tell us that

$$\frac{dt}{d\lambda} = -\frac{k_{0i} z^2}{h(z)}; \quad \frac{d\vec{x}}{d\lambda} = \vec{k}_i z^2; \quad -\frac{h(z)}{z^2} \left( \frac{dt}{d\lambda} \right)^2 + \frac{1}{z^2 h(z)} \left( \frac{dz}{d\lambda} \right)^2 + \frac{1}{z^2} \left( \frac{d\vec{x}}{d\lambda} \right)^2 = 0, \tag{4.44}$$

where the subscript $i$ indicates the different geodesics that end up at boundary points $(t_i, x_i)$. We remind the reader that these geodesics can move along the *complexified* $(t, \vec{x}, z)$ manifold. Nevertheless, we will consider geodesics that originate at real $z$ since the contour of integration originally runs along real $z$. For bulk to boundary propagators, the geodesic runs from a bulk point at an initial real value of $z$, which we denote by $z_r$, to the boundary, which is at $z = 0$ and so we are interested in geodesics whose imaginary part again becomes zero when $Re(z) = 0$.

Using the last equation to solve for $\frac{dz}{d\lambda}$ we find that

$$\frac{dz}{d\lambda} = -z^2 \left( k_{0i}^2 - \vec{k}_i^2 h(z) \right)^{\frac{1}{2}}. \tag{4.45}$$

The $z$-equation can be integrated to yield

$$\lambda = \frac{1}{z\sqrt{k_{0i}^2 - \vec{k}_i^2}} \, {}_2F_1\left(\frac{1}{2}, -\frac{1}{d}; \frac{d-1}{d}; \frac{\vec{k}_i^2}{\vec{k}_i^2 - k_{0i}^2}\frac{z^d}{z_0^d}\right) - f_0, \tag{4.46}$$

where the constant $f_0$ is set according to the convention that the affine parameter is 0 for $z = z_r$. Near the boundary, we have

$$\lambda \xrightarrow[z \to 0]{} \frac{1}{\sqrt{k_{0i}^2 - k_i^2}z}, \tag{4.47}$$

and so $\lambda$ tends to $\infty$ near the boundary.

Since the geodesic must reach the boundary at a real value of $z = 0$, the allowed geodesics must satisfy

$$k_{0i}^2 - \vec{k}_i^2 > 0. \tag{4.48}$$

We now proceed to prove that null geodesics obey (4.43). Our proof proceeds in two steps.

1 First we show that the equation (4.44) where the derivatives of $t$ and $\vec{x}$ need to be integrated along the curve (4.46) to obtain the full displacement $\delta t$ and $\delta \vec{x}$ can also be integrated along the real $z$-axis by considering the equations

$$\frac{dt}{dz} = \frac{1}{\frac{dz}{d\lambda}}\frac{dt}{d\lambda} = \frac{1}{h(z)\sqrt{1 - \vec{c}_i^2 h(z)}}; \qquad \frac{d\vec{x}}{d\lambda} = \frac{1}{\frac{dz}{d\lambda}}\frac{d\vec{x}}{d\lambda} = \frac{\vec{c}_i}{\sqrt{1 - \vec{c}_i^2 h(z)}}, \tag{4.49}$$

where $\vec{c}_i = \frac{\vec{k}_i}{k_{i0}}$. What we need to prove here is that $\frac{dz}{d\lambda} \neq 0$ at any point between the trajectory of the original geodesic and the real $z$-axis. If so, then we can deform the integration contour of the equations (4.49) from the original geodesic to the real $z$-axis.

2 Then we show that along the real $z$ axis, the condition (4.43) is satisfied.

We will assume, throughout this analysis that $\text{Im}(\vec{c}_i) \neq 0$. This is the generic case, and our proof is easily generalized to the special case where $\text{Im}(\vec{c}_i) = 0$.

To prove property 1, we first note that along real $z$-axis, $\text{Im}(\sqrt{1 - \vec{c}_i^2 h(z)})$ cannot change signs. This quantity can only change sign if, at some point on the real $z$-axis, we have $1 - \vec{c}_i^2 h(z) > 0$ as a real number. But this is impossible since $h(z) \in R$ but $\text{Im}(\vec{c}_i^2) \neq 0$.

Now consider the *family* of geodesics with the same value of $\vec{k}_i$ and $k_{i0}$ but starting at different initial values of the $z$-coordinate: $0 < z(0) < z_r$. These geodesics *cannot* intersect the original geodesic that starts at $z(0) = z_r$ because the derivative along the curve is purely a function of $z$ so a unique curve passes through each complex value of $z$ where $\frac{dz}{d\lambda} \neq 0$. These geodesics also cannot intersect the $z$-axis. This is because if the geodesic intersects the $z$-axis, then $\frac{d\text{Im}(z)}{d\lambda} = \text{Im}\left(-z^2\sqrt{(k_{0i}^2 - \vec{k}_i^2) - \vec{k}_i^2(h(z) - 1)}\right)$ must have different signs at $z = z_r$ and

the point where it returns to the $z$-axis. However, by (4.48) and a simple extension of the argument above, $\frac{d\text{Im}(z)}{d\lambda}$, keeps a fixed sign for real $z$. Therefore these geodesics must stay *between* the real $z$-axis and the trajectory of the original geodesic that starts at $z_r$. If we additionally assume that the geodesic curve varies continuously as the initial starting point varies then it follows that *all points* in the complex $z$-plane between the original geodesic and the real $z$-axis can be reached by varying $z(0)$. But since all geodesics terminate at the boundary, this means that $\frac{dz}{d\lambda} \neq 0$ for any point between the original geodesic and the real $z$-axis.

This implies that to obtain the displacement $\delta\vec{x}$ and $\delta t$ we may integrate their derivatives, given by (4.49) along the real $z$-axis. Note that this immediately allows us to obtain explicit formulas for $\delta t$ and $\delta \vec{x}$ by explicit integration

$$
\delta t_i = \frac{z F_1 \left( \frac{1}{d}; \frac{1}{2}, 1; 1 + \frac{1}{d}; \frac{\bar{c}_i^2 z^d}{\bar{c}_i^2 - 1}, z^d \right)}{\sqrt{1 - \bar{c}_i^2}},
$$

$$
\delta \vec{x}_i = \vec{c}_i \frac{z \, {}_2F_1 \left( \frac{1}{2}, \frac{1}{d}; 1 + \frac{1}{d}; \frac{\bar{c}_i^2 z^d}{\bar{c}_i^2 - 1} \right)}{\sqrt{1 - \bar{c}_i^2}},
$$

(4.50)

where $F_1$ is the "Appell F-function".

Now we show property 2. First we perform a rotation in the transverse directions so that $\vec{k}_i = (k_i, 0, \ldots 0)$ and therefore $\vec{c}_i = (c, 0 \ldots 0)$ with $c = \frac{k_i}{k_{i0}}$. Second, for convenience, we consider the case where $\text{Im}(c) > 0$ so that $\text{Im}(\frac{1}{\sqrt{1-c^2 h(z)}}) > 0$ and $\text{Im}(\frac{c}{\sqrt{1-c^2 h(z)}}) > 0$. The other cases can be treated by trivially changing some signs below.

We define

$$
D(d, z) = \text{Im} \left[ \frac{c}{\sqrt{1 - c^2 h(z)}} - \frac{1}{h(z)\sqrt{1 - c^2 h(z)}} \right].
$$

(4.51)

We noting that, through some simple algebra, if $D(d, z)$ vanishes for $0 < h < 1$, this can only happen at

$$
h = \frac{-1 + \text{Re}(c)}{|c|^2 - \text{Re}(c)}.
$$

(4.52)

Since $D(d, z) > 0$ at $z = 0$ ($h = 1$) this means that $D(d, z)$ is positive near $z = 0$ and can cross the real axis *at most once* between the boundary and $z_r$.

To prove (4.43), we only need to integrate $D(d, z)$ from the position of the contour to the boundary. However, the property of $D(z)$ above tells us that (4.43) will be implied if we prove that

$$
H(d) = \int_0^{z_0} D(d, z) > 0.
$$

(4.53)

We will prove this as follows. First, we will show that once $H(d)$ becomes positive, it remains positive as we increase $d$. Then we will check that for $d = 2$, the integral is positive which proves that it is positive for all $d$.

First, we note that

$$\frac{\partial D(d, z)}{\partial d} = \frac{\partial D(d, z)}{\partial z} \frac{z}{d} \log(\frac{z}{z_0}). \tag{4.54}$$

Therefore

$$d\frac{\partial}{\partial d} H(d) = d\frac{\partial}{\partial d} \int_0^{z_0} D(d, z) = \int_0^{z_0} \frac{\partial D(d, z)}{\partial z} z \log(\frac{z}{z_0}) = - \int_0^{z_0} D(d, z) \log(\frac{z}{z_0}) - H(d). \tag{4.55}$$

The boundary terms in the integration by parts vanish because the log vanishes at $z = z_0$, while at the boundary $z = 0$.

The differential equation above can be written as

$$\frac{\partial}{\partial d} dH(d) = - \int_0^{z_0} D(d, z) \log(\frac{z}{z_0}). \tag{4.56}$$

Now by the assumption about $D$ above we also have that

$$\int_0^{z_0} D(d, z) \log(\frac{z}{z_0}) < 0. \tag{4.57}$$

since $\log(\frac{z}{z_0}) < 0$ and moreover $|\log(\frac{z}{z_0})|$ becomes larger as we go closer to $z = 0$. Therefore in the integral above $\log(z)$ weights the positive section of $D$ with a weight that is larger in magnitude than the weight for the section where $D$ is negative.

Turning now to $H(2)$ this can be analytically computed using the formulas above to be $H(2) = \text{Im}(\log(1 + c))$. Recall that we are considering the case where $\text{Im}(c) > 0$ so that clearly $H(2) > 0$. Therefore $H(d) > 0, \forall d \geq 2$. The result (4.43) now follows immediately.

**Analyticity of correlators in the domain $\eta_i \in \mathcal{F}$**    The analyticity of correlators in the required domain can now be proved. We let $\eta_i$ be the *imaginary part* of the displacement of the boundary points from each other and let $(\delta t_i, \delta \vec{x}_i)$ be the displacement in time and space from the point where the contour may be pinched in the bulk. Then, we see that the imaginary parts of these displacements are given by

$$(\text{Im}(\delta t_1), \text{Im}(\delta x_1)), (\text{Im}(\delta t_1), \text{Im}(\delta x_1)) + \eta_1, \ldots (\text{Im}(\delta t_1), \text{Im}(\delta x_1)) + \eta_1 + \ldots \eta_{n-1}. \tag{4.58}$$

However starting from a null or spacelike vector and adding future directed timelike vectors, it is *not* possible to obtain a configuration of spacelike vectors whose spatial parts are *not* on one-side of some codimension 1 hyperplane. So the singularity cannot be pinched by boundary points whose imaginary displacements are in the future timelike direction.

In fact, starting with the initial contour that runs from $Im(t) = 0$ to $Im(t) = -\beta$ and along $Im(\vec{x}) = 0$, we can deform the contour as shown in Figure 9. In the $Im(t), Im(\vec{x})$ plane, the contour follows a *causal path* that tracks all the boundary points. Such a path must exist since the imaginary displacement between each point and the next point in the Wightman correlator is timelike and future directed.

On this contour of integration, it is clear that there are *no* singularities that remain even in the integrand. this is because a singularity can only arise when one of the boundary points

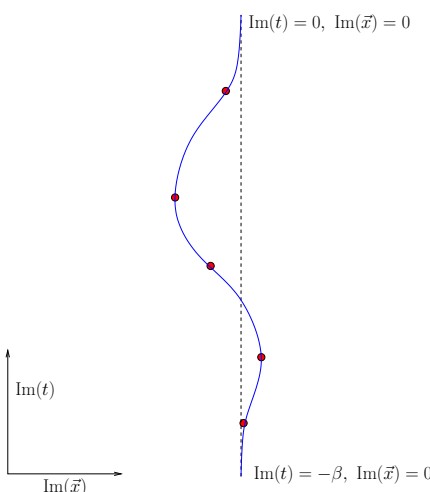

$\text{Im}(t) = 0, \ \text{Im}(\vec{x}) = 0$

$\text{Im}(t)$

$\text{Im}(\vec{x})$

$\text{Im}(t) = -\beta, \ \text{Im}(\vec{x}) = 0$

**Figure 9**: *A deformation of the integration contour that explicitly avoids all singularities for a contact Witten diagram. The boundary points are displayed as red dots. The original contour is the dashed line along $\text{Im}(\vec{x}) = 0$. The final deformed contour is the blue line. The red dots are the imaginary coordinates of the boundary insertions.*

is separated from a point on the contour by a spacelike imaginary displacement. However, in Figure 9 the imaginary displacement between every boundary point and every point on the contour is timelike.

It is also clear why the proof breaks down if the condition $\beta e_i - \sum \eta_i \in \mathcal{V}^+$ is not met. Since the boundary is identified in Euclidean time, it is possible for geodesics to go both "forward" and "backward" in imaginary time. For points that are outside the diamond of analyticity, what may seem like a timelike displacement $\eta_i \in \mathcal{V}^+$ may nevertheless be reached by a light ray going in the "wrong" direction in time. See Figure 10. In this situation, it is clear that the proof of the previous section does not hold.

### 4.2.2 Exchange interactions

We now turn to exchange interactions. Let $G(t, x, z, t', x', z')$ be the bulk-bulk propagator. Then exchange interactions are given by summing Witten diagrams which yield integrals of the form

$$W(t_i, x_i) = \int \prod_{i_1=1}^{n_1} K_{\Delta_i}(t_i, \vec{x}_{i_1}, t, \vec{x}, z) G(t, \vec{x}, z, t', \vec{x}', z')$$

$$\times \prod_{i_2=1}^{n_2} K_{\Delta_{i_2}}(t_{i_2}, \vec{x}_{i_2}, t', \vec{x}', z') G(t', \vec{x}', z', t'', \vec{x}'', z'') \dots dt dt' d^{d-1}\vec{x} d^{d-1}\vec{x}' \frac{dz dz'}{z^{d+1}(z')^{d+1}} \cdots$$

$$(4.59)$$

Now the singularities of the bulk to bulk propagator are also along the light-cone, and so we need to repeat the analysis of the previous subsection for bulk-bulk propagators.

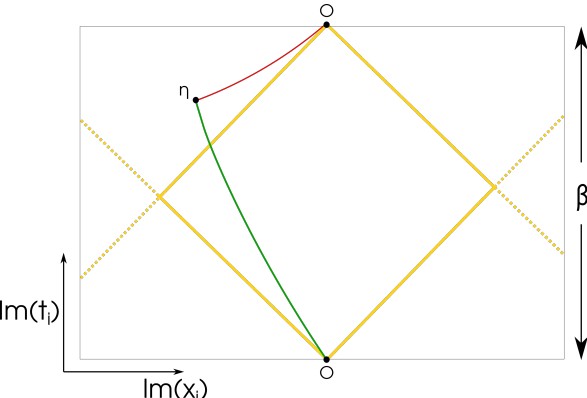

**Figure 10**: *It seems like the vector $\eta$'s imaginary coordinates are "timelike" separated from $\mathcal{O}$. But if we take into account the periodic identification of Euclidean time, it is also possible to reach $\eta$ from $\mathcal{O}$ via a "spacelike" imaginary displacement.*

Some parts of the previous argument go through. For example, if the bulk-bulk propagator starts and ends at real values of $z$, then the displacement of $\delta\vec{x}$ and $\delta t$ between the two end-points of the bulk-bulk propagator can again be obtained by integrating the geodesic equations along the real $z$ axis. For such geodesics, the condition (4.48) may not hold. Instead, such geodesics are separated by a point on the real $z$-axis, where the sign of $\mathrm{Im}(\frac{dz}{d\lambda})$ changes sign. This is because if this quantity has one sign at one endpoint (as the geodesic departs the real-$z$-axis), it must have the opposite sign at the other endpoint (as the geodesic returns to the real-$z$-axis). This point occurs at the unique value of $h(z)$ for $0 < z < 1$ where

$$h(z_t) = \frac{\mathrm{Im}(k_0^2)}{\mathrm{Im}(k_i^2)} \tag{4.60}$$

provided that we also have $\mathrm{Re}(k_0^2) - h(z_t)\mathrm{Re}(k_i^2) > 0$. Consider a geodesic that starts at $z_r$ and terminates at $z_r'$. Once again we consider geodesics with the same value of $k_0$ and $\vec{k}_i$ that start between $z_r$ and $z_t$. Since these geodesics cannot intersect the original geodesic they must intersect the $z$ axis at some point between $z_t$ and $z_r'$. Proceeding this way, by starting with different initial conditions, we can "fill up" the entire region between the real $z$-axis and original geodesic with other geodesics. This means that there are no points where $\frac{dz}{d\lambda}$ vanishes in the region between the trajectory of the geodesic and the real $z$-axis.

However, it is *not* true that for the bulk-bulk propagator, the vector $(\mathrm{Im}(\delta t), \mathrm{Im}(\delta\vec{x}))$ must be spacelike. To consider a trivial counter-example, consider a geodesic that propagates from a value of $z$ close to the horizon to another value of $z$ close to the horizon. In the near-horizon region, we can make $|\mathrm{Im}\left(\frac{1}{h(z)\sqrt{1-c^2h(z)}}\right)| > |\mathrm{Im}\frac{c}{\sqrt{1-c^2h(z)}}|$ and therefore we can easily achieve $|\mathrm{Im}(\delta t)|^2 > \mathrm{Im}(\delta\vec{x}_i) \cdot \mathrm{Im}(\delta\vec{x}_i)$.

This means that the proof of the previous subsection, that applied to contact Witten diagrams is not immediately applicable.

This does not mean that exchange diagrams violate our bound. For example, in some cases, such as the BTZ black hole, exchange Witten diagrams can be reduced to sums of contact diagrams by extending the techniques of [38]. Then the proof of the previous subsection indirectly implies that exchange diagrams also have the correct analytic properties. However, for the more general case, we have not yet been able to find an appropriate proof that exchange Witten diagrams lead to boundary correlators obeying our bound. We leave the question of the analytic properties of exchange Witten diagrams as an open problem.

## 5  Conclusions

In this paper, we considered a novel limit of correlation functions of Wightman correlators in a relativistic quantum field theory at finite temperature, where the spatial momenta of the insertions became large while their frequencies remained finite. We showed, using very general properties of quantum field theories, that the correlator was bounded by the exponential of a specific geometric term: the radius of the smallest sphere that could contain the non-planar polygon of the momenta in units of the temperature.

This bound applies to correlators of any local operator. If one focuses on the case of correlators of elementary fields in a perturbative quantum field theory, then generically the perturbative expansion produces terms that could saturate this bound at high enough loop order for an arbitrary $n$-point correlator.

Since holographic theories are strongly coupled, one might have suspected that they would always saturate this bound. This would be consistent with the general intuition that, at strong coupling, all processes that are allowed at some order in perturbation theory are indeed realized. However, at least at the level of the leading two-point function we find that holographic correlators in $d > 2$ fail to saturate the bound.

This paper only initiates the study of this interesting limit, and there are several open questions that deserve further attention. For instance, while the bound holds for asymptotically large momenta does it also necessarily hold in large $N$ theories for momenta that are large compared to the temperature but small compared to $N$? What happens in holographic theories? If the bound is obeyed at large-$N$, then at what order in bulk perturbation theory is the bound saturated? And how does bulk perturbation theory in momentum space reorganize itself so that dangerous terms that could violate the bound cancel? Is the characteristic under-saturation of the bound at strong coupling in higher dimensions a distinctive feature of holographic theories and a characteristic of the horizon?

It appears that the answer to these questions may lie in a closer study of the analytic properties of holographic thermal correlation functions. It also seems important to develop better techniques to actually compute such correlators, which would allow these formal predictions to be compared with concrete calculations.

### Acknowledgments

We are grateful to Soumyadeep Chaudhuri, Ben Freivogel, Chandan Jana, Shiraz Minwalla, Vladimir Rosenhaus, Bo Sundborg, Nico Wintergerst, Amos Yarom and especially to R. Loganayagam for several helpful discussions. S.R. is partially supported by a Swarnajayanti Fellowship of the Department of Science and Technology (India). K.P. would like to thank ICTS, Bangalore for hospitality. This research was supported in part by the International Centre for Theoretical Sciences (ICTS) during a visit for participating in the program - AdS/CFT at 20 and Beyond (Code: ICTS/adscft20/05). The work of SB is supported by the Knut and Alice Wallenberg Foundation under grant 113410212.

## A    Thermal perturbation theory

In this appendix, we review the elements of thermal perturbation theory. Our analysis applies to any perturbative quantum field theory and is applied in the main text, both to holographic correlators, and to weakly-coupled theories. We first describe a canonical formulation of thermal perturbation theory, and then describe a diagrammatic formulation that naturally arises from the Schwinger-Keldysh representation. The material covered here is standard, but we include it here for the sake of completeness and also because it is somewhat difficult, in the extant literature, to find a clear and concise description of the rules to carry out perturbation theory for relativistic field theories.

### A.1    Canonical formulation

We are interested in evaluating

$$\mathrm{Tr}(e^{-\beta H}\phi(t_1, x_1)\ldots\phi(t_n, x_n)). \tag{A.1}$$

At some point of time, $\tau$ we split the Hamiltonian into a free and an interacting part

$$H = H_0[\tau] + H_I[\tau]. \tag{A.2}$$

Here $H_I$ is the interaction Hamiltonian evaluated at real time $\tau$ and $H_0$ is the "free Hamiltonian", also evaluated at time $\tau$. Note that both $H_0$ and $H_I$ depend on the time we choose to make this split, $\tau$, although this dependence on $\tau$ must eventually drop out. Below, whenever $H_I$ is evaluated at time $\tau$, we will suppress this dependence to lighten the notation.

Now consider

$$T(z) = e^{zH_0}e^{-zH}. \tag{A.3}$$

This satisfies

$$\begin{aligned}
T'(z) &= e^{zH_0}H_0e^{-zH_0}e^{zH_0}e^{-zH} - e^{zH_0}He^{-zH_0}e^{zH_0}e^{-zH} \\
&= -e^{zH_0}H_I e^{-zH_0}T(z).
\end{aligned} \tag{A.4}$$

The solution to this is just

$$T(\beta) = \mathcal{T}_c e^{-\int_0^\beta H_I(\tau - iz)dz}. \tag{A.5}$$

where

$$H_I(\tau - iz) = e^{zH_0} H_I e^{-zH_0}, \tag{A.6}$$

and $\mathcal{T}_c$ denotes a contour-ordering sign, where the contour moves down in imaginary time from $\text{Im}(t) = 0$ to $\text{Im}(t) = -\beta$. In the expression above larger values of $z$ are placed to the left. More explicitly, we have

$$T(\beta) = \sum_n \frac{(-1)^n}{n!} \int \mathcal{T}_c\{H_I(\tau - iz_1)\dots H_I(\tau - iz_n)\}dz_1 \dots dz_n. \tag{A.7}$$

Therefore we have

$$e^{-\beta H} = e^{-\beta H_0} T(\beta). \tag{A.8}$$

We can expand the interaction Hamiltonian as a sum of terms with various frequencies (as measured by the free Hamiltonian). If we then write

$$H_I(t) = \int_{-\infty}^{\infty} H_I(\omega) e^{-i\omega t} \frac{d\omega}{2\pi}, \tag{A.9}$$

then we have

$$H_I(t - iz) = \int_{-\infty}^{\infty} e^{-i\omega t - z\omega} H_I(\omega) \frac{d\omega}{2\pi}, \tag{A.10}$$

and

$$T(\beta) = \sum_n (-1)^n \int \prod \frac{d\omega_i}{2\pi} \int_0^\beta dz_1 \int_0^{z_1} dz_2 \dots \int_0^{z_{n-1}} dz_n e^{-i\sum \omega_i(\tau - iz_i)} H(\omega_1) \dots H(\omega_n). \tag{A.11}$$

We now turn to the real-time part of the correlator. Using standard arguments we have

$$\phi(t_i, x_i) = \overline{\mathcal{T}}\left[e^{i\int_\tau^t H_I(x)dx}\right] \phi_I(t, x_i) \mathcal{T}\left[e^{-i\int_\tau^t H_I(x)dx}\right], \tag{A.12}$$

where $\phi_I(t, x_i)$ is the interaction-picture operator at time $t$.

With a little algebra this can be written as

$$\phi(t_i, x_i) = \sum_{N=0}^{\infty} i^N \int_\tau^t dt_N \dots \int_\tau^{t_2} dt_1 [H_I(t_1), [H_I(t_2) \dots [H_I(t_N), \phi_I(t, x_i)] \dots]]. \tag{A.13}$$

*Combining* (A.13) *and* (A.7) *we immediately obtain a perturbative expansion for* (A.1).

For consistency, we would like to see the following two effects emerge from the expressions above

1. Although we have suppressed this dependence, in fact both the free Hamiltonian and the interaction Hamiltonian depend on the time at which we make the split, $\tau$, and correspondingly $\tau$ also appears in the lower limit of the integral.

2. Second, the correlator above should be time-translationally invariant. So if we shift $t_i \to t_i + x$, the correlator should not change.

This is obvious in the original expression (A.1). However, in perturbation theory this appears to be a little puzzling. To see the puzzle, let us suppress the separate time-dependence and instead consider a single operator $C(t)$. The generalization to operators at different times will be given later, and will be obvious.

Therefore, we consider the expression $\mathrm{Tr}(e^{-\beta H}C(t))$. We will expand this out to second order in perturbation theory to check the two consistency properties above. To second order we have

$$
\begin{aligned}
T(\beta) = 1 &- \int H_I(\omega)e^{-i\omega\tau}\frac{1 - e^{-\beta\omega}}{\omega}\frac{d\omega}{2\pi} \\
&+ \int H_I(\omega)H_I(\omega')e^{-i(\omega+\omega')\tau}\Big[\frac{1 - e^{-\beta\omega}}{\omega\omega'} + \frac{e^{-\beta(\omega+\omega')} - 1}{(\omega+\omega')\omega'}\Big]\frac{d\omega}{2\pi}\frac{d\omega'}{2\pi}.
\end{aligned}
\tag{A.14}
$$

Further, we write the interaction picture operator as

$$
C_I(t) = \int C_I(\omega)e^{-i\omega t}\frac{d\omega}{2\pi}.
\tag{A.15}
$$

Inserting this into the nested commutators above yields an expression for the Heisenberg-picture operator, which we will use below.

Before we turn to the general structure of the perturbative expansion we work out the first order terms and the quadratic terms explicitly. The reader may skip these explicit calculations if she is interested only in the results.

**First order terms:**

The first order terms are

$$
\int F\frac{d\omega}{2\pi}\frac{d\omega'}{2\pi},
$$

where

$$
F = -\mathrm{Tr}e^{-\beta H_0}\Big[H_I(\omega)\frac{1 - e^{-\beta\omega}}{\omega}C_I(\omega')e^{-i\omega't}e^{-i\omega\tau} - \frac{1}{\omega}[H_I(\omega), C_I(\omega')]e^{-i\omega't}\big(e^{-i\omega t} - e^{-i\omega\tau}\big)\Big].
\tag{A.16}
$$

In general, we expect this correlator to have support for all values with $\omega + \omega' = 0$. However, this is puzzling, since in some of the terms above, we appear to get a non-zero dependence on both $t$ and $\tau$.

This can be resolved by imposing the *KMS condition*.

$$
\begin{aligned}
\mathrm{Tr}\left(e^{-\beta H_0}H_I(\omega)C_I(\omega')\right) &= e^{\beta\omega}\mathrm{Tr}\left(H_I(\omega)e^{-\beta H_0}C_I(\omega')\right) = e^{\beta\omega}\mathrm{Tr}\left(e^{-\beta H_0}C_I(\omega')H_I(\omega)\right) \\
&= e^{\beta\omega}\mathrm{Tr}\left(e^{-\beta H_0}\big(H_I(\omega)C_I(\omega') - [H_I(\omega), C_I(\omega')]\big)\right).
\end{aligned}
\tag{A.17}
$$

In particular this means that

$$
(1 - e^{-\beta\omega})\mathrm{Tr}\left(e^{-\beta H_0}H_I(\omega)C_I(\omega')\right) = \mathrm{Tr}\left(e^{-\beta H_0}[H_I(\omega), C_I(\omega')]\right).
\tag{A.18}
$$

Therefore, we have

$$F = \frac{-1}{\omega} \text{Tr}(e^{-\beta H_0} [H'_I(\omega), C_I(\omega')]) e^{-i\omega' t} e^{-i\omega t}, \quad \omega, \omega' \neq 0. \tag{A.19}$$

Since the trace is proportional to $\delta(\omega + \omega')$, in this form, it is clear that the correlator is independent of both $\tau$ and $t$.

However, the contribution above is *not* the full contribution to the correlator since in writing the final expression for $F(\omega, \omega')$ we divided by $\omega$. This is not allowed at $\omega = 0$. In particular, if the thermal expectation of $H_I(\omega) C_I(\omega')$ has a term proportional to $\delta(\omega) \delta(\omega')$. This term is not cancelled off by the KMS condition. However, this term is also manifestly independent of $\tau$ and $t$ so the puzzle above does not arise here. We will return to these correction terms below.

**Second order terms:**

Now let us consider the second order terms. We need to include the second order term from $T(\beta)$ the second order term from the real-time evolution, and the product of the first order terms. Therefore the full expression we need to consider is as follows

$$S = \int \frac{d\omega_1 d\omega_2 d\omega_3}{(2\pi)^3} \text{Tr} e^{-\beta H_0} \Big[ T_1 + T_2 + R \Big]$$

$$T_1 = e^{-i(\omega_1 + \omega_2)\tau - i\omega_3 t} \int_0^\beta dz_1 \int_0^{z_1} dz_2 e^{-z_1 \omega_1 - z_2 \omega_2} H_I(\omega_1) H_I(\omega_2) C_I(\omega_3)$$

$$T_2 = -i e^{-i\omega_1 \tau - i\omega_3 t} \int_0^\beta dz_1 e^{-z_1 \omega_1} \int_\tau^t dt_2 e^{-i\omega_2 t_2} H_I(\omega_1) [H_I(\omega_2), C_I(\omega_3)] \tag{A.20}$$

$$R = -\int_\tau^t dt_2 \int_\tau^{t_2} dt_1 e^{-i\omega_1 t_2 - i\omega_2 t_2 - i\omega_3 t} [H_I(\omega_1), [H_I(\omega_2), C_I(\omega_3)]].$$

We again consider the case where $\omega_1 \neq 0, \omega_2 \neq 0, \omega_3 \neq 0$. For the term denoted by $T_1$ above we need to use the KMS relations twice. This yields

$$\text{Tr}(e^{-\beta H_0} H_I(\omega_1) H_I(\omega_2) C_I(\omega_3)) = \frac{1}{(1 - e^{-\beta\omega_1})(1 - e^{-\beta\omega_2})} \text{Tr}(e^{-\beta H_0} [H_I(\omega_2), [H_I(\omega_1), C_I(\omega_3)]])$$

$$- \frac{e^{-\beta\omega_3}}{(1 - e^{-\beta\omega_1})(1 - e^{-\beta\omega_3})} \text{Tr}(e^{-\beta H_0} [C_I(\omega_3), [H_I(\omega_1), H_I(\omega_2)]]). \tag{A.21}$$

We can put both these terms in the form of the commutator that appears in the real-time expression by using the Jacobi identity for the second expression

$$[C_I(\omega_3), [H_I(\omega_1), H_I(\omega_2)]] = -[H_I(\omega_1), [H_I(\omega_2), C_I(\omega_3)]] + [H_I(\omega_2), [H_I(\omega_1), C_I(\omega_3)]]. \tag{A.22}$$

After these steps, we find that

$$T_1 = \frac{\left(\omega_1\left(e^{-\beta\omega_2}-1\right)+\omega_2\left(e^{\beta\omega_1}-1\right)\right)e^{-i\tau(\omega_1+\omega_2)}}{\omega_1\omega_2\left(e^{\beta\omega_1}-1\right)\left(e^{\beta\omega_2}-1\right)\left(e^{\beta\omega_3}-1\right)\left(\omega_1+\omega_2\right)}e^{-i\omega_3 t}$$
$$\times\left[\left(e^{\beta(\omega_2+\omega_3)}-1\right)\mathrm{Tr}(e^{-\beta H_0}[H_I(\omega_2),[H_I(\omega_1),C_I(\omega_3)]])\right.$$
$$\left.-\left(e^{\beta\omega_2}-1\right)\mathrm{Tr}(e^{-\beta H_0}[H_I(\omega_1),[H_I(\omega_2),C_I(\omega_3)]])\right]. \tag{A.23}$$

We also find that

$$T_2 = \frac{e^{-i\omega_3 t-i\tau\omega_1}}{\omega_1\omega_2}\left(e^{-it\omega_2}-e^{-i\tau\omega_2}\right)\mathrm{Tr}(e^{-\beta H_0}[H_I(\omega_1),[H_I(\omega_2),C_I(w3)]]), \tag{A.24}$$

whereas the real-time term is given by

$$R = -e^{-i\omega_3 t}\frac{\left((\omega_1+\omega_2)e^{-i(t\omega_2+\tau\omega_1)}-\omega_2 e^{-it(\omega_1+\omega_2)}+\omega_1\left(-e^{-i\tau(\omega_1+\omega_2)}\right)\right)}{\omega_1\omega_2(\omega_1+\omega_2)}$$
$$\times\mathrm{Tr}(e^{-\beta H_0}[H_I(\omega_1),[H_I(\omega_2),C_I(\omega_3)]]). \tag{A.25}$$

Upon adding these terms, and noting that within the integral we can switch the dummy variables $\omega_2\leftrightarrow\omega_1$ we find that the full quadratic term in the integrand for $\omega_i\neq 0$ is

$$T_1+T_2+R = \frac{e^{-i\omega_3 t}}{\omega_1\omega_2}\mathrm{Tr}(e^{-\beta H_0}[H_I(\omega_1),[H_I(\omega_2),C_I(\omega_3)]]\Bigg[$$
$$\frac{e^{-(\beta+i\tau)(\omega_1+\omega_2)}}{\left(e^{\beta\omega_1}-1\right)\left(e^{\beta\omega_2}-1\right)\left(e^{\beta\omega_3}-1\right)\left(\omega_1+\omega_2\right)}\times\left(\omega_1\left(e^{\beta(2(\omega_1+\omega_2)+\omega_3)}-e^{\beta(\omega_1+\omega_2)}+e^{\beta\omega_1}\right)\right.$$
$$\left.-(\omega_1+\omega_2)e^{\beta(2\omega_1+\omega_2+\omega_3)}+\omega_2\left(e^{\beta(\omega_1+2\omega_2)}-e^{\beta\omega_2}+e^{\beta(\omega_1+\omega_2+\omega_3)}-e^{2\beta(\omega_1+\omega_2)}+e^{\beta(2\omega_1+\omega_2)}\right)\right)$$
$$+\frac{\omega_2 e^{-it(\omega_1+\omega_2)}+\omega_1 e^{-i\tau(\omega_1+\omega_2)}}{\omega_1+\omega_2}+e^{-i\tau(\omega_1+\omega_2)}+\dots, \tag{A.26}$$

where ... indicates terms that either integrate to 0 or contribute only when one of the $\omega_i$'s is 0.

However, recall that the thermal trace has support only on $\omega_3=\omega_1+\omega_2$. This is because the trace can be evaluated in any basis, including the basis of eigenstates of $H_0$ in which the total $H_0$-eigenvalue of the insertion inside the trace must vanish. Imposing this condition, we find a tremendous simplification in the expression above and the full quadratic term becomes

$$T_1+T_2+R = \frac{1}{\omega_1^2+\omega_1\omega_2}\mathrm{Tr}(e^{-\beta H_0}[H_I(\omega_1),[H_I(\omega_2),C_I(\omega_3)]])+\dots. \tag{A.27}$$

Even though the integral above is over three variables it is understood that when we evaluate the trace, this will force the constraint $\omega_3=\omega_1+\omega_2$.

**Result: General structure of the perturbative expansion**

From the examples above, we arrive at the following general structure of the perturbative expansion. In the perturbative expansion, there are two terms that are multiplied by phases dependent linearly on $\tau$. One term comes from the expansion of $T(\beta)$. the other term comes from the *lower* limit of the time-integrals. The two example calculations above show that these two terms cancel with each other.

Since the full amplitude cannot depend on $\tau$ in any manner, this cancellation *must* continue to all orders. Therefore, *for generic frequencies of the operators* that appear in the perturbative expansion, the only term that can survive from the multiple time-integrals comes from the *upper limit* of integration. The contribution from the lower-limit of integration cancels with the contribution from (A.11) for generic values of $\omega_i$.

However, this is *not* the full contribution to the correlation function. As pointed out below (A.19) and in the discussion leading to (A.27), there may be terms in the correlation function that, in the space of frequencies of the insertions, appear on surfaces of codimension 1 or higher. These terms are, by themselves, independent of $\tau$ and, in general, our argument that they cancel does not apply.

For instance, in (A.11) we may expect to get a finite contribution to the correlator from frequencies that satisfy $\sum \omega_i = 0$. We can quantify this contribution by extracting the part in the product of the interaction Hamiltonians that is proportional a delta function in the $\omega_n$ (A.11)

$$(-1)^n \int_0^\beta dz_1 \int_0^{z_1} dz_2 \ldots \int_0^{z_{n-1}} dz_n e^{-i \sum \omega_i (\tau - i z_i)} H(\omega_1) \ldots H(\omega_n) = \mathcal{Z}_1(\omega_i) 2\pi \delta(\sum \omega_i) + \ldots,$$
(A.28)

where ... indicates terms that contribute for generic values of $\omega_i$. Then we set

$$(1 + Z_1) = \sum_n \int \prod \frac{d\omega_i}{2\pi} \mathcal{Z}_1(\omega_i) 2\pi \delta(\sum \omega_i).$$
(A.29)

However, one may also have contributions that appear from terms where the sum of frequencies in (A.7) cancels with a frequency from the lower-limit of real-time integration from the commutators. We write this contribution as $Z_2$ and we will quantify it when we turn to the Schwinger-Keldysh formalism.

This leads to the following general result At $n^{\text{th}}$ order in perturbation theory we find that

$$\text{Tr}(e^{-\beta H} C(t))$$
$$= \sum_n \int \prod_{i=1}^{n+1} \frac{d\omega_i}{2\pi} \text{Tr}(e^{-\beta H_0}(1 + Z_1) g(\omega_i)[H_I(\omega_1), \ldots [H_I(\omega_n), C_I(\omega_{n+1})]] e^{i \sum \omega_i t} + Z_2),$$
(A.30)

where the factors $Z_1$ and $Z_2$ are discussed above. Even though the integral runs over $n + 1$ variables, the thermal trace yields $\delta(\omega_1 + \ldots \omega_{n+1})$ and therefore all functions depend only

on $n$ variables. The function $g(\omega_i)$ comes from the upper limit of time-integration and is therefore given by

$$g(\omega_i) = \frac{(-1)^n}{\prod_{k=1}^{n}\sum_{j=1}^{k}\omega_j} = \frac{(-1)^n}{\omega_1(\omega_1+\omega_2)\dots(\omega_1+\dots\omega_n)}. \tag{A.31}$$

The alert reader might worry that (A.30) that $H_0$, $H_I$ and also $C_I$ are all implicitly dependent on $\tau$. However, consider making the split between the free and the interacting part at a different time $\tau' = \tau + x$. Then we note immediately that

$$H_0' = e^{iHx}H_0e^{-iHx}; \quad H_I' = e^{iHx}H_Ie^{-iHx}. \tag{A.32}$$

However, denoting the *Heisenberg picture* operator by $C_H$, we have

$$\begin{aligned}
C_I'(\omega) &= \int_{-\infty}^{\infty} e^{iH_0't}C_H(\tau')e^{-iH_0't}e^{i\omega t}dt \\
&= \int_{-\infty}^{\infty} \left(e^{iHx}e^{iH_0t}e^{-iHx}\right)\left(e^{iHx}C_H(\tau)e^{-iHx}\right)\left(e^{iHx}e^{-iH_0t}e^{-iHx}\right) \\
&= e^{iHx}C_I(\omega)e^{-iHx}.
\end{aligned} \tag{A.33}$$

We now see, using the cyclicity of the trace that the factors of $e^{-iHx}$ all cancel on the right hand side of (A.30) so this correlator does not depend on $\tau$ as expected.

The result (A.30) can be easily generalized to evaluate a Wightman function that involves insertions at different times. We find that

$$\mathrm{Tr}(e^{-\beta H}C(t_1)\dots C(t_n)) = \sum \int \left(\prod_{j,l}\frac{d\omega_l^j}{2\pi}\right)\prod_j g(\omega_l^j)e^{i\sum_{j,l}\omega_l^j t_j}\mathrm{Tr}\Big(e^{-\beta H_0}(1+Z_1) \tag{A.34}$$

$$\times [H_I(\omega_1^1)\dots[H_I(\omega_{s_1}^1),C_I(\omega_{s_1+1}^1]\dots]\dots[H_I(\omega_1^n),\dots[H_I(\omega_{s_n}^n),C_I(\omega_{s_n+1}^n)]\dots] + Z_2\Big).$$

Physically, this formula can be understood as follows. Consider taking $\tau \to -\infty$. This means that the split between the free and the interaction Hamiltonian is performed at $t = -\infty$. Then, if we proceed *naively* we might imagine that

1. By means of a suitable turning on/off function for the interaction we can make the full Hamiltonian coincide with the free Hamiltonian at $\tau = -\infty$.

2. In the time-integrals that arise from the Dyson expansion, we can ignore all the terms that arise from the lower limit of integration.

These steps are too naive because in the thermal case, as we adiabatically turn on the interaction we may heat or cool the state or change it in some other manner. This is the explanation for the term $Z_1$ above. In fact, if the system does not *thermalize* effectively, then some contributions from early times may remain important even at late times and this is the physical explanation for the term $Z_2$ above.

If we choose the interaction-term carefully so that it does not change the temperature of the system then $Z_1$ may just be a numerical factor that will cancel when we compute thermal expectation values since it will also appear in the partition function. If the system thermalizes effectively then $Z_2 = 0$ but this is a very subtle issue as we discuss below.

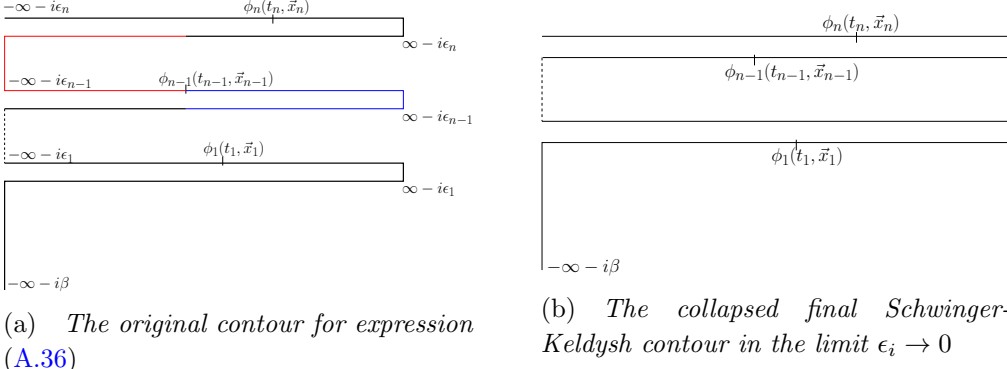

(a)   *The original contour for expression (A.36)*

(b)   *The collapsed final Schwinger-Keldysh contour in the limit $\epsilon_i \to 0$*

**Figure 11**: The Schwinger-Keldysh contour

## A.2   Schwinger-Keldysh formalism

In this section we will briefly describe the Schwinger-Keldysh formalism, which yields a diagrammatic approach to computing thermal Wightman functions in relativistic field theories. In the process we will also clarify the functions $Z_1$ and $Z_2$ above. Consider again the thermal expectation value (A.1). We now give small negative imaginary parts to the time coordinates $t_i \to t_i - i\epsilon_i$ so that $\epsilon_1 > \epsilon_2 > \ldots \epsilon_n$. At the end of the calculation we will take $\epsilon_i \to 0$. Then we can represent all the points on a time-contour that runs from $-\infty - i\epsilon_n \to \infty - i\epsilon_n$, snakes back to $-\infty - i\epsilon_n$, then moves down in imaginary time to $-\infty - i\epsilon_{n-1}$ goes to $+\infty - i\epsilon_{n-1}$ and so on. At the end the contour moves in imaginary time and ends up at $t = -\infty - i\beta$ as shown in Figure 11a.

To write an expression for the correlator using this contour, we adopt the notation

$$U_{Ii}(t_1, t_2) \equiv e^{-i\int_{t_1}^{t_2} H_{Ii}(\tilde{t}-i\epsilon_i)d\tilde{t}}. \tag{A.35}$$

Then we can write, using the analysis of the previous section,

$$\begin{aligned}
&\text{Tr}(e^{-\beta H}\phi(t_1, \vec{x}_1)\phi(t_2, \vec{x}_2)\ldots\phi(t_n, \vec{x}_n)) \\
&= \text{Tr}(e^{-\beta H_0}\mathcal{T}_c\Big\{T(\beta)U_{I1}(-\infty, \infty)\phi_I(t_1, \vec{x}_1)U_{I1}(\infty, -\infty) \\
&U_{I2}(-\infty, \infty)\phi_I(t_2, \vec{x}_2)U_{I2}(\infty, -\infty)\ldots U_{In}(-\infty, \infty)\phi_I(t_n, \vec{x}_n)U_{In}(\infty, -\infty)\Big\}),
\end{aligned} \tag{A.36}$$

where $\mathcal{T}_c$ denotes ordering *along* the contour.

Now, in the limit where the $\epsilon_i \to 0$, note that the expression (A.36) has multiple redundancies since parts of the various $U_I$ operators cancel with each other. In Figure 11a, for instance, the red parts of the contour cancel and so do the blue parts, leaving only the thick black part. This allows us to collapse the contour of 11a to 11b. Even the contour of 11b has redundancies. However, it is convenient to retain these redundancies in order to obtain easy Feynman rules.

Note that the final number of horizontal legs in the collapsed contour of Figure 11b is $\tilde{n}$ where $\tilde{n} = n$ if $n$ is even and $\tilde{n} = n + 1$ if $n$ is odd. This is because the contour must return to $-\infty$ before descending to $-\infty - i\beta$.

To obtain the Feynman rules, we now introduce $\tilde{n} + 1$ types of fields, corresponding to the factors $\tilde{n}$ horizontal legs of the contour and the single vertical leg. We can define a contraction[4] of these $\tilde{n}+1$ fields that can be evaluated in terms of ordinary interaction-picture fields as follows

$$
D_{ij}(t_1, \vec{x}_1, t_2, \vec{x}_2) = \frac{1}{Z_0} \times \begin{cases} \mathrm{Tr}(e^{-\beta H_0} \mathcal{T} \phi_I(t_1, \vec{x}_1)\phi_I(t_2, \vec{x}_2)) & i = j \text{ and } n - i \text{ even}, \\ \mathrm{Tr}(e^{-\beta H_0} \overline{\mathcal{T}} \phi_I(t_1, \vec{x}_1)\phi_I(t_2, \vec{x}_2)) & i = j \text{ and } n - i \text{ odd}, \\ \mathrm{Tr}(e^{-\beta H_0} \phi_I(t_1, \vec{x}_1)\phi_I(t_2, \vec{x}_2)) & i > j, \\ \mathrm{Tr}(e^{-\beta H_0} \phi_I(t_2, \vec{x}_2)\phi_I(t_1, \vec{x}_1)) & i < j. \end{cases} \tag{A.37}
$$

The first two lines above correspond to time-ordered and anti-time-ordered thermal expectation values whereas the last two lines correspond to Wightman functions where the field that appears later on the contour is placed first. These two-point functions can be calculated using (3.5) and (3.7). For instance, the time-ordered propagators are,

$$
\frac{1}{Z_0} \mathrm{Tr}(e^{-\beta H_0} \mathcal{T} \phi_I(t_1, \vec{x}_1)\phi_I(t_2, \vec{x}_2)) = \int \frac{d^{d-1}\vec{k}}{2(2\pi)^d \omega_k} e^{i\vec{k}\cdot(\vec{x}_1 - \vec{x}_2)} \frac{e^{-i\omega_k|t_1 - t_2|} + e^{-\beta\omega_k} e^{i\omega_k|t_1 - t_2|}}{1 - e^{-\beta\omega_k}}. \tag{A.38}
$$

We can Fourier transform the propagator to obtain a momentum-space expression.

$$
D_{ij}(k_1)(2\pi)^d \delta(k_1 + k_2) = \int d^d x_1 d^d x_2 D_{ij}(x_1, x_2) e^{-ik_1 \cdot x_1 - ik_2 \cdot x_2}, \tag{A.39}
$$

and these are the expressions listed in (3.10).

The second feature that will appear when we expand out (A.36) using Wick's theorem is that the interaction Hamiltonian appears with a positive sign for odd legs of the contour and a negative sign for even legs of the contour. These rules also apply to the vertical segment where we take the value of $t$ to be complex. If both legs are on the vertical segment, then the propagator is the Euclidean two-point function and if one leg is on the vertical segment and another is on a horizontal segment then the propagator is the analytically continued Wightman function.

Therefore, in the end, when we expand out (A.36) and take $\epsilon_i \to 0$, we get the following Feynman rules

1. There are $\tilde{n}+1$-types of interaction vertices. Of these $\tilde{n}$ correspond to the different $H_I(t)$ on the horizontal parts of the contour. The $0^{\mathrm{th}}$ vertex corresponds to the interaction

---

[4]The contraction can be defined, as usual, as the difference of the contour-ordered product and the "normal ordered product". However, the "normal ordered product" must be defined, by making a Bogoliubov transform of the creation and annihilation operators so that its *thermal* expectation value vanishes. This is explained in [39]

vertex on the vertical part of the contour. The $j^{\text{th}}$ vertex connects only fields of type $i$ to each other and has a coefficient $(-1)^{n-j}(-i)$. The $0^{\text{th}}$ vertex comes has a coefficient $(-1)$.

2. All interaction vertices on the horizontal parts of the contour are integrated from time $-\infty$ to $\infty$ and over all space.

3. The interaction vertex on the vertical part of the contour is integrated in Euclidean time from $[0, \beta]$ and over all space.

4. There are $(\tilde{n}+1)^2$-types of propagators that connect fields of type $i$ to fields of type $j$ as given in (A.37)

5. The external legs correspond to fields of type $1 \ldots n$.

### More on the vertical part

The vertical part of the contour corresponds to a very subtle term. First note that the Feynman-diagram expansion yields terms where interaction Hamiltonians from the vertical part only contract with each other through a Euclidean propagator. These terms contribute a disconnected set of graphs that are not connected to the external points. This is an overall numerical prefactor that is clearly just $\frac{\text{Tr}\left(e^{-\beta H}\right)}{\text{Tr}\left(e^{-\beta H_0}\right)}$.

Now, as we take the vertical part of the contour in real time to $-\infty$, we may expect that the mixed propagators that connect the vertical and horizontal part die off due to the Riemann-Lebesgue lemma. However, this does not always happen because some terms in the Feynman diagram may continue to contribute at $t = -\infty$. This is in contrast to the situation in perturbation theory about the vacuum, where by evolving infinitely along a slightly imaginary direction we can project out all contributions except those corresponding to the vacuum. It is this contribution from the vertical part of the contour that leads to the factors $Z_1$ and $Z_2$ in (A.30). This subtlety has been discussed in the thermal field theory literature and we refer the reader to [19, 23, 40] for more details. In our calculations in the main text, we will *not* include the contribution of the vertical part of the contour. We do not believe that this will materially affect our results, but we leave a more detailed discussion of these effects to a later study.

## B  Interactions in the BTZ black hole

In this appendix, we provide some more details of holographic contact Witten diagrams for the BTZ black hole. We consider a four-point interaction between scalar field with dimensions $\Delta_i$. In the BTZ black-hole we can explicitly compute the bulk-boundary propagators, but our analysis here is entirely *complementary* to the analysis in the main text. The alert reader may have noticed that in using (4.41) in the main text, we did not need to use the condition of linear dependence of the normals. This condition is only meaningful if less than $d+1$-singularities collide since otherwise it is met trivially. In this Appendix, we will see the

relevance of this condition for a four-point function and we will not need to use (4.42) at all in this Appendix.

We will consider the *Euclidean, planar* BTZ black hole with metric

$$ds^2 = (r^2 - r_+^2)d\tau^2 + \frac{1}{r^2 - r_+^2}dr^2 + r^2 d\phi^2. \tag{B.1}$$

Here the temperature is given by $\beta = T^{-1} = \frac{2\pi}{r_+}$. The coordinate $r$ here is related to the coordinate $z$ used in the main text through $r = \frac{1}{z}$.

The bulk to boundary propagator in this geometry between a boundary point $(\tau_i, \phi_i)$ and a bulk point $(\tau, r, \phi)$ can then be found to be [41]

$$K_{\Delta_i} = \frac{N_{\Delta_i}}{\left(\cosh(r_+(\phi_i - \phi))\frac{r}{r_+} - \cos(r_+(\tau_i - \tau))\sqrt{\frac{r^2}{r_+^2} - 1}\right)^{\Delta_i}}. \tag{B.2}$$

A contact Witten diagram in Euclidean space can then be calculated to be

$$G(t_i, \phi_i) = \int r dr d\tau d\phi \prod_{i=1}^{4} \frac{N_{\Delta_i}}{\left(\cosh(r_+(\phi_i - \phi))\frac{r}{r_+} - \cos(r_+(\tau_i - \tau))\sqrt{\frac{r^2}{r_+^2} - 1}\right)^{\Delta_i}}, \tag{B.3}$$

where $\Delta_i$ are the dimensions of the fields that participate in the interaction and $N_{\Delta_i}$ is a normalization that will be irrelevant for us.

To get the Lorentzian Wightman function with arguments extended in imaginary time, we can write $\tau_i = it_i + \delta_i$ and the ordering in the Wightman correlator is set by the ordering of the $\delta_i$. Similarly, we can extend the transverse coordinates in the imaginary direction through $\phi_i = x_i + i\epsilon_i$ in the imaginary direction inside the integral expression (B.3). Let us order the $\delta_i$ so that $\delta_1 < \delta_2 < \delta_3 < \delta_4$. Without loss of generality, we set $\delta_1 = \epsilon_1 = 0$; this just corresponds to setting the first point in the four-point function to the origin, which can be done by a translation. Then, to check the analyticity properties in position space, we need to check the following property: *Provided (i) $\delta_i - \delta_{i-1} > 0$ and (ii) $|\delta_i - \delta_{i-1}| > |\epsilon_i - \epsilon_{i-1}|$ and (iii) $|\beta - \delta_4| > |\epsilon_4|$, the integral should not have any singularities.*

Notice that the *integrand* in (B.3) then has singularities whenever

$$S_i = \cosh(r_+(x_i + i\epsilon_i - \phi))\frac{r}{r_+} - \cos(r_+(it_i + \delta_i - \tau))\sqrt{\frac{r^2}{r_+^2} - 1} = 0. \tag{B.4}$$

We see that the first bulk-boundary propagator cannot encounter a singularity. But the other three bulk-boundary propagators can encounter singularities and in principle, either two or three singularities can collide at a point on the integration contour. We now show that this cannot happen in such a way as to satisfy (4.41).

**Two singularities colliding** We will now prove that that contour cannot be pinched by the meeting of any *two* singularities. Notice that when $S_i = 0$, we have

$$
\begin{aligned}
\frac{\partial S_i}{\partial r} &= \frac{1}{r_+} \left( \cosh(r_+(x_1 + i\epsilon_1 - \phi)) - \frac{\frac{r}{r_+}}{\sqrt{(\frac{r}{r_+})^2 - 1}} \cos(r_+(it_i + \delta_i - \tau)) \right) \\
&= \frac{-1}{r_+} \cosh(r_+(x_1 + i\epsilon_1 - \phi)) \frac{1}{(\frac{r}{r_+})^2 - 1}.
\end{aligned}
\tag{B.5}
$$

If the contour is pinched between two singularities, and if $r < \infty$, we would find, by demanding linear dependence of the derivatives, that

$$
\begin{aligned}
\frac{\sinh(r_+(x_1 + i\epsilon_1 - \phi))}{\sinh(r_+(x_2 + i\epsilon_2 - \phi))} &= \frac{\sin(r_+(it_1 + \delta_1 - \tau))}{\sin(r_+(it_2 + \delta_2 - \tau))} = \frac{\cosh(r_+(x_1 + i\epsilon_1 - \phi))}{\cosh(r_+(x_2 + i\epsilon_2 - \phi))} \\
&= \frac{\cos(r_+(it_1 + \delta_1 - \tau))}{\cos(r_+(it_2 + \delta_2 - \tau))}.
\end{aligned}
\tag{B.6}
$$

These conditions require the points to be either coincident or else separated by $\Delta\epsilon = \Delta\delta = \frac{2\pi}{r_+}$. The second case, requires the imaginary shift to be larger than $\beta$, whereas the first case involves a coincident singularity.

**Three singularities colliding** We may also consider the case, where three $S_i$ vanish simultaneously for some $r < \infty$. In this case, imposing the linear dependence of derivatives implies that, for some constants $\gamma_1, \gamma_2$ we have

$$
\begin{aligned}
\cosh(r_+(x_3 + i\epsilon_3 - \phi)) &= \gamma_1 \cosh(r_+(x_1 + i\epsilon_1 - \phi)) + \gamma_2 \cosh(r_+(x_2 + i\epsilon_2 - \phi)); \\
\sinh(r_+(x_3 + i\epsilon_3 - \phi)) &= \gamma_1 \sinh(r_+(x_1 + i\epsilon_1 - \phi)) + \gamma_2 \sinh(r_+(x_2 + i\epsilon_2 - \phi)); \\
\cos(r_+(it_3 + \delta_3 - \tau)) &= \gamma_1 \cos(r_+(it_1 + \delta_1 - \tau)) + \gamma_2 \cos(r_+(it_2 + \delta_2 - \tau)); \\
\sin(r_+(it_3 + \delta_3 - \tau)) &= \gamma_1 \sin(r_+(it_1 + \delta_1 - \tau)) + \gamma_2 \sin(r_+(it_2 + \delta_2 - \tau)).
\end{aligned}
\tag{B.7}
$$

Using the fact that $\cosh^2 x - \sinh^2 x = \cos^2 x + \sin^2 x = 1$ we see that the equations above imply that we must have

$$
\cosh(r_+(x_1 - x_2 + i\epsilon_1 - i\epsilon_2)) = \cos(r_+(i(t_1 - t_2) + (\delta_1 - \delta_2))),
\tag{B.8}
$$

which immediately tells us that (by writing the cos as a cosh and equating the imaginary part of the argument and excluding the case where the shift in imaginary coordinates is larger than $\beta$) that $|\epsilon_1 - \epsilon_2| = |\delta_1 - \delta_2|$ at the singularity. Namely that the singularity cannot occur if we keep the difference of $\delta$'s larger than the difference of $\epsilon$.

For the four-point function, we cannot have the situation where four or more singularities coincide in the interior. This situation is relevant for higher-point functions and in such a case, the linear dependence of the normals can be met trivially. We now need to impose the additional condition, (4.42) imposed in the text: the contour cannot be pinched if the imaginary part of the boundary points are on one side of a hyperplane since by deforming the contour, we can remove the singularities. However, it is interesting that this condition does not seem to be required for Witten diagrams with a small number of external legs.

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
