# Peer review of "A Bound on Thermal Relativistic Correlators at Large Spacelike Momenta"

_SciPost Physics_

## Round 1 · Referee Report · Anonymous (Referee 1) · 2019-9-12

Strengths

1- A general result about thermal Wightman correlators in relativistic QFTs.

2- The proof is clear and straightforward.

3- Very well written.

4- Techniques used in section 2, 3, and 4 can be of importance in other contexts.

Weaknesses

1- Discussions about weakly coupled scalar field theories and holographic theories are long and technical. So, section 3 and 4 are somewhat of a distraction from the main point of the paper which is very general.

Report

This paper studies thermal Wightman correlators at large spatial momenta but finite frequencies in relativistic QFTs. In this particular limit, the authors derive a geometric bound on the decay rate of thermal Wightman correlators. The bound follows directly from the analytic structure of position space correlators in relativistic unitary QFTs and hence this bound should be universal. Then the authors show that the bound is saturated in weakly coupled scalar field theories at sufficiently high loop orders. On the other hand, the bound is not saturated in holographic theories above $d=2$. This paper is well-written overall and I would happily recommend this article for publication, once the following comments/suggestions/questions have been taken into account.

Requested changes

I have following comments/suggestions/questions:

(1) Is there a position space version of the bound?

(2) It is known that the position space correlators are analytic on a larger domain than considered in this paper. Can the full domain of analyticity improve the bound?

(3) Equation (2.7) should be edited to avoid confusion: $|\vec{y}_1-\vec{y}_2|+\cdots+|\vec{y}_{n-1}-\vec{y}_n|+|\vec{y}_n-\vec{y}_1|\le \beta$.

(4) What happened to the factor $e^{i \vec{k}_i . \vec{x}_i}$ in the equation before (2.9)?

(5) Many of the equations in section 2.3 should be edited. For example see (2.13)-(2.18) where dot products are omitted (such as $\vec{P}_i . \vec{a}_i$, $\vec{k}_i . \vec{y}_i$ and some others.)

(6) I have a general question about thermal Wightman correlators at large spatial momenta. Let us consider the two-point function results from section 3 or 4. On physical grounds, one would expect that at large momenta the two-point function should asymptote to the vacuum two-point function. In fact, the position space (equal time) thermal two-point function for the holographic theory as well as the free scalar field theory approaches the vacuum result at short distances. Where does that intuition break down? Probably this should be addressed in the paper as well.

  • validity: high
  • significance: high
  • originality: top
  • clarity: high
  • formatting: good
  • grammar: excellent

Author:  Suvrat Raju  on 2019-11-24  [id 653]

(in reply to Report 1 on 2019-09-12)

We would like to thank the referee for their useful comments. We have made the following updates to the paper to address the referee's points. (Below, we follow the same numbering as the referee does in the referee-report.)

(1) The equivalent statement of the bound in position space is that the correlators are analytic in the domain F discussed in the paper.

(2) We have added a comment about a possibly larger domain of analyticity in a footnote on page 6.

(3) We have updated these equations following the recommendations of the referee.

(4) The typo has been corrected and the factor restored.

(5) The dot products have been restored in section 2.3.

(6) We have added a comment about the large spacelike momentum limit approaching vacuum correlators at the end of the paragraph which is under equation (1.2).

---

## Editorial Decision

resubmitted